# Breast Cancer: How Hippotherapy Bridges the Gap between Healing and Recovery—A Randomized Controlled Clinical Trial

**DOI:** 10.3390/cancers15041317

**Published:** 2023-02-19

**Authors:** Hélène Viruega, Corinne Galy, Célia Loriette, Stéphane Jacquot, Jean Louis Houpeau, Manuel Gaviria

**Affiliations:** 1Equiphoria Institute, Combo Besso Rouges Parets, 48500 La Canourgue, France; 2Clinical Neurosciences, Alliance Equiphoria, 4 Res du Sabot, 48500 La Canourgue, France; 3General and Homeopathic Practitioner, 2 rue Canton, 34090 Montpellier, France; 4Department of Oncology and Radiotherapy, Clinique Clémentville, 25 rue de Clémentville, 34070 Montpellier, France; 5Department of Senology, Clinique Clémentville, 25 rue de Clémentville, 34070 Montpellier, France

**Keywords:** breast cancer, integrative oncology, post-treatment supportive care, rehabilitation, hippotherapy, wholeness, personalized care, life reconstruction, empowerment, quality of life

## Abstract

**Simple Summary:**

Breast cancer is the most diagnosed women’s cancer, and has a high survival rate nowadays. Because cancer is a systemic disease characterized by a variable course, heterogeneity and unequal environmental inputs, disparities in the person’s future are the norm. Despite enormous progress in the early accurate detection of breast cancer, and its treatments becoming more effective/precise, life reconstruction is well beyond the current care path. It requires comprehensive cross-sectoral approaches between different knowledge areas, and deeper consideration of the challenges the patients have to deal with. The psychological and social sciences must be integrated with the physiological sciences to build a robust patient-centered healthcare practice. We demonstrate, through this clinical trial, the therapeutic relevance of hippotherapy, a one-health approach, as a key initial stage after cancer diagnosis and treatment to foster recovery. Furthermore, hippotherapy has a strong impact on cancer treatments’ efficiency and reconstruction of the individuals’ shattered life and their ecosystem. This work reveals a layer of complexity that needs to be broadly considered.

**Abstract:**

Background: Breast cancer is the most diagnosed women’s cancer, and has a high survival rate. Despite great progress in detection and treatment, life reconstruction requires comprehensive cross-sectoral approaches between different disciplines and deeper consideration of the patient’s challenges. Hippotherapy is an emerging specialized rehabilitation approach, performed by accredited health professionals and equine specialists, on specially trained horses via their movement, activating major paths for physical, mental, psychic and social reinforcement, and is synergistic to rehabilitative and supportive care. Methods: We conducted a randomized open, prospective, two-armed, controlled trial on the effectiveness of hippotherapy versus conventional supportive care on adult women with a diagnosis of breast cancer, after the period of primary treatment (surgery, chemotherapy, radiotherapy). The 6-month program included, in the treated group, an initial 1-week daily hippotherapy session, followed by three short 2-day sessions with an interval of 2 months between each, where the patients received conventional supportive care. The control group received 6 months of conventional supportive care. The primary end point was quality of life. Cognitive performances, fatigue, anxiety, depression, and body image were the secondary end points. Measurements were done through self-reported questionnaires. Results: We observed statistical differences in the evolution of the measured parameters over time between the two groups. The hippotherapy group showed a much faster, favorable and continuous improvement until the end of the program for each function assessed. The most striking improvements were observed in global quality of life, and fatigue, while breast cancer-specific quality of life, cognitive performance, anxiety and depression and body image showed a less marked but still statistically significant difference at the final post-treatment evaluation. Conclusions: We demonstrate the therapeutic relevance of hippotherapy, a one-health approach, as a key initial stage after cancer diagnosis and treatment to foster recovery. Furthermore, hippotherapy has a strong impact on cancer treatments’ efficiency and reconstruction of patient’s life and ecosystem. This work reveals a layer of complexity that needs to be broadly considered. Trial registration: ClincalTrials.gov NCT04350398 accessed on 1 January 2022. Registered 17 April 2020, retrospectively registered; French Clinical Trials in Cancer Register RECF3818. Registered 18 March 2019, retrospectively registered.

## 1. Introduction

Breast cancer remains a major worldwide public health issue [1]. It is the most frequent women’s cancer and the most common cause of cancer death within this population group [2,3]. Concomitantly with significant progress in diagnostic accuracy, the rapid development of cutting-edge personalized treatment options for breast cancer is a key feature of the modern oncology workflow, tremendously improving the healing and survival rates [2,4]. Nevertheless, biological healing always reveals further events of major clinical relevance that are compulsory to deal with [5,6]. The medical interpretation and management of a patient’s global post-cancer health status often pose considerable challenges to the modern healthcare system [7], and rely on deeply personalized, multilayered comprehension and care, well beyond the processes of basic physiologic cellular and molecular reset. Indeed, following an event of such violence in both its symbolism and in its management, all the human spheres are massively impacted. The person, who has completely relied on medical intervention and treatments for healing, has disowned their body, and must find a way to fully recover their sense of self, in its wholeness. After this personal tsunami, creating a vital impulse and a new direction that fits one’s intimate needs and wants is imperative for one to move on with their life.

Hippotherapy can tackle this challenge by providing a self-committed, efficient, non-intrusive, custom-made framework of the patient’s physical, mental, emotional and spiritual spheres [8,9,10,11,12,13,14]. We have, therefore, developed a wholeness-through-hippotherapy program that allows patients to move beyond recovery, and that warrants novel forms of adjustments in order to nurture the more standard outcomes related to the attenuation of functional deficits [15,16,17]. Hippotherapy is an emerging specialized rehabilitation approach, performed on a specially trained horse, via its movement at a walk, by a team of accredited health professionals (e.g., physicians, psychologists, physical therapists, occupational therapists, and psychomotor therapists among others), and equine specialist who lead the horse [9,11,14,18,19,20,21,22,23]. The horse’s walking movements are biomechanically similar to the human movements when walking, and yield comparable micro-adjustments of the patient’s postural balance, gross and fine motor abilities and reinforcement of sensorimotor control [24]. Additionally, through multimodal inputs (sensory, exteroceptive, proprioceptive, interoceptive, and emotional), hippotherapy has a direct action on the individual’s cognitive potentials and emotional regulation, through the interactions of several neural networks [10,12,22,25,26,27,28,29]. During hippotherapy, the specific execution and repetition of a task are key elements of learning/strengthening/promoting a function, and robust rehabilitation and reconstruction processes [12,22,26,27,28,29,30]. It has been reported, for example, that emotionally empowering multimodal interventions, such as hippotherapy, can provide patients who are in a late post-stroke phase with life-changing experiences that can have a profound physical and psychological impact [14,31,32]. In this study, we compared the different impacts—physical, cognitive, emotional—of breast cancer diagnosis and treatments on two cohorts of patients during the first year of post-cancer diagnosis and treatments. The cohort consisted of women diagnosed with stage I to IIIA breast cancer (T1-T3, N0-N2 and M0), who had completed primary treatment (surgery, radiotherapy, chemotherapy), with the exception of hormonal therapy. On the treated group, patients underwent a hippotherapy program at the Equiphoria Institute (www.equiphoria.com, accessed on 1 January 2023), in addition to a supportive care program consisting of physical therapy, psychotherapy, nutrition and dietetics, algology, homeopathy, acupuncture, sexology, addictology, and physical activity (provided by the ‘Montpellier Institut du Sein’—MIS, Care and Support Unit: www.le-mis.fr, accessed on 1 January 2023), according to the needs and wishes of the patients. In the control group, patients underwent only the supportive care path in the MIS study site, according to their needs and wishes. The results revealed a remarkable improvement in the treated group, on all spheres evaluated.

To our knowledge, this is an unprecedented effort in Europe for a comprehensive groundbreaking empowerment of women diagnosed with and treated for breast cancer during a critical phase of resumption of a normal life [33]. Here, we describe our approach and discuss the results of our clinical trial, conducted from October 2016 to August 2022. In summary, this work provides a large, clinically relevant resource for the management of women treated for breast cancer, for whom reliable healthcare programs beyond the current track are urgently needed.

## 2. Materials and Methods

### 2.1. Cohort Selection

A randomized, controlled, data evaluator-blind clinical trial (#NCT04350398) was conducted in accordance with the Declaration of Helsinki and with good clinical practice guidelines, following approval by the ethics committee of Montpellier, France (CPP Sud-Mediterranée IV authorization n° 161201 on 27 January 2017), and the French Agency for the Safety of Health Products (ANSM clinical trial authorization n°2016122600049 on 26 December 2016). In France, the Jardé’s law defines Research Involving the Human Person (RIPH) as research organized and carried out on healthy or sick volunteers, with a view to developing biological or medical knowledge, and which aims to evaluate the safety and efficacy of the proposed healthcare solution. RIPH may only be carried out after a favorable opinion from a university hospital ethical committee (CPP), drawn at random at a national level from an IT platform set up by the French health authorities. This new procedure replaces the former procedure of ethical evaluation by local university committees. All patients provided written informed consent.

A total of 82 patients were recruited in the trial. Among the 82 participants, 66 completed the trial (hippotherapy *n* = 35; controls *n* = 31), with a final dropout rate of 22% and 16%, respectively. The first patient was enrolled on 20 April 2017 and the last on 30 October 2021. All patients were recruited from the MIS. Participating patients were adult women between 26 and 78 years of age with a histologically confirmed diagnosis of breast cancer (T1-T3, N0-N2 and M0) undergoing or completing treatment (surgery and/or chemotherapy and/or radiotherapy), with a WHO performance index [34] of 0 to 2, who had consulted a physician of the MIS Care and Support Unit during the cancer treatment period, able to give written informed consent, able to complete questionnaires, affiliated to a social security scheme, and who had provided a certificate of no contraindication to hippotherapy issued by their physician. Patients were excluded if they had a history of malignancies in the last 5 years, a history of chronic fatigue syndrome, concomitant and uncontrolled severe degenerative or chronic diseases, clinically significant cognitive impairment or dementia, pregnancy and breastfeeding, a history of therapeutic horse riding or hippotherapy during the last 6 months, and if the patient was participating in another biomedical research project or was in an exclusion period. Patients were divided into two groups, hippotherapy and controls, which were equally balanced for primary cancer treatment and age, using a balanced fixed randomization protocol (see Table 1).

### 2.2. Statistics and Reproducibility

We used the RStudio (2022.02.1; Boston, MA, USA) language and environment to achieve statistical computing and graphic creation [35]. The sample size was determined to detect a difference with a power of 80%, using a two-sample *t*-test at a two-sided significance level of 5%. The minimum reasonably significant QoL difference was Δ = 10 points, according to other similar studies [36,37,38]. The variance in total QoL in four recently published studies [38,39,40,41] is, respectively, σ = 9.1 (*n* = 75), σ = 16.4 (*n* = 25), σ = 13.9 (*n* = 19), and σ = 17.8 (*n* = 33). We chose the study of Vardar [40], which seemed to be the most adapted to the characteristics of our population, and which showed an important dispersion in a small cohort. The standard deviation used to calculate the sample size was, therefore, σ = 13.9. A total of 70 patients in the randomized arms (35 per arm) were required (moreover, we calculated an extra 20% to prevent a reduction of power due to potential additional dropouts). Six semi-quantitative, self-administered questionnaires were completed by the patients (see Appendix A Case Report Form in the linked document: HippoBreastCa_CRF_VE_31012017.pdf):The European Organization for Research and Treatment of Cancer Quality of Life Questionnaire (EORTC QLQ-C30 and EORTC QLQ-BR23)The Functional Assessment of Cancer Therapy-Cognitive Function (FACT-Cog V3)The Multidimensional Fatigue Inventory (MFI-20)The Hospital Anxiety and Depression (HADS)The Body Image Scale (BIS)

Categorical variables were analyzed through Chi-squared statistics. We tested the evolution of the scores of each questionnaire across time in the two groups and we compared them. We also tested the evolution of the scores obtained at the first post-treatment evaluation and at the final evaluation, after baseline correction for each participant. Baseline correction was achieved by subtracting the score obtained at Q1 from the score obtained at Q2 and Q3 for each subset score and for each participant. This allowed us to set all the participants to a score of zero and erase the variability of the baseline states. In addition, by doing so, we were able to compare the differences in score changes between the two groups. When baseline correction is performed, a positive score observed at Q2 or Q3 means an increase of the score compared to Q1, and a negative score observed at Q2 or Q3 implies a decrease of the score compared to Q1. Baseline correction was achievable, since the baseline values were comparable in the two groups (no significant differences). Table 2 shows the summary scores (see Section 2.4.2 for the computing formula of EORTC QLQ-C30 and EORTC QLQ-BR23) of the different clinical tests, as well as the individual scores for each item before the beginning of the program for the hippotherapy and the control group, and the statistical result of the two-by-two comparison. We tested the normality of the data using a Shapiro–Wilk test and none of the data followed a normal distribution. This was predictable, as none of the scores followed a continuous distribution. In order to be as rigorous as possible, considering the non-continuous distribution, the relatively small sample sizes and the results of the Shapiro–Wilk test, we decided to use nonparametric statistics. To test the evolution of scores across time for each treatment, we performed a Friedmann test, followed by a two-sided paired Wilcoxon test to achieve ad hoc post-test analysis. For testing the difference between the two treatments across time, we tested the data through a robust between–within participants ANOVA, using the WRS2 package adapted to non-parametric data [42,43]. Then, two-sided unpaired Wilcoxon tests were used to perform ad hoc post-tests. For each statistical test, the *p*-value, the degree of freedom (df), if available, and its statistic (F-value for ANOVA, Friedmann Chi-squared for Friedmann, V and W statistics for paired and unpaired Wilcoxon, respectively) are reported.

All the statistics were performed blind, as the analyst was not aware of the treatment applied to each statistical group. Given the estimated power of the study (80%), the post hoc results were considered statistically robust, and therefore reliable. Statistical significance is described with the following code on the figures: (ns) non-significant, (.) close to significance (between 0.08 and 0.05), (*) *p* < 0.05, (**) *p* < 0.01, (***) *p* < 0.001. The threshold of statistical significance was set to *p* = 0.05. We chose SEM to be displayed on the figures for graphical reasons. However, the standard deviations (SD) are provided in Table 2 and Appendix C Table A1 and Table A2 for more accuracy.

### 2.3. Treatment

Two study groups were set up: a treated group receiving the hippotherapy treatment combined with a supportive care program (except during the hippotherapy days, to avoid an additive effect of hippotherapy), and a control group receiving only the supportive care program. We tested the response to treatments through a series of questionnaires (see below). The timeline of the trial is described in Figure 1. Three measurement time points were defined: before the healthcare program (baseline evaluation), after the first one-week session of the healthcare program (first post-treatment evaluation), and at the end of the six-month healthcare program (final evaluation). The hippotherapy sessions took place at the Equiphoria Institute, a specialized rehabilitation-through-hippotherapy clinic in the South of France. Each patient was assisted by at least one or two professionals when by the horse, and two or three professionals when on the horse, in order to guarantee their safety. The patient was systematically equipped with a special multi-loop belt and a helmet when on the horse.

*Hippotherapy:* The 6 month program consisted of an initial 1-week daily hippotherapy session (5 consecutive days) by the end of the initial cancer treatment. This initial hippotherapy period was completed by three 2-day daily sessions with an interval of 2 months between each. During the remaining time, the patients followed the same supportive care as the control group (see below). The hippotherapy exercises on the horse (1 h) and by the horse (2 h) consisted of different performed exercises focused on diverse entangled health functioning processes: reinforcement of global postural balance and fine-tuning of postural responses (eyes open and closed); strengthening of different muscle groups; working on upper limbs’ fine motor skills and coordination; work on smoothness of movement and psycho-corporal relaxation; reinforcement of the body schema and body image [44]; breathing techniques and visualizations; attention to body/emotional feelings; support for attention, concentration, working memory, and executive functions; reinforcement of the notions of success, pleasure, and disinhibition; social and cognitive flexibility; reinforcement of communication skills; stimulation of proactive postures, involvement and motivation; self-esteem and self-transcendence; work on values, meaning, appropriation of symbols; the management of post-traumatic stress disorder, the grieving process, and the creation of a new life course [45]; linguistic activity of creating meaning (describing, explaining, reappropriating, reinventing, and projecting oneself). The hippotherapy protocol was substantially the same for every patient. A routine set of exercises was used according to the needs of each patient on the horse (breathing exercises; visualizations; body scan; relaxation exercises; active movement of different body segments and exercises to improve body contextualization; therapeutic vaulting) and by the horse (choice of the horse and grooming; various walking courses with the horse; round pen work; blindfolded guidance or grooming; work at liberty). The intensity of the exercises was fully tailored and depended on the patient’s clinical heterogeneity, fatigability, and confounding factors. It is challenging to determine the minimum effective quantity of a required rehabilitation or supportive care. Scientific evidence is lacking, and studies have biases of several kinds. To date, the effectiveness of many such techniques has not been systematically demonstrated. A hippotherapy session lasts one hour per day, during which between 3000 and 4500 contractions of each postural muscle are sequentially realized in a background mode (horse at a walk), in parallel to other requests (fine motor skills, cognitive elaboration, and psychic work), which is well beyond what a conventional rehabilitation or supportive care session allows. Given the intensity of each session, which mobilizes the individual in his/her entirety (e.g., somatic, sensory, cognitive, emotional and motivational, and psychic spheres), and relying on the enriched environment brought by hippotherapy, we have, most of the time, noticed a remarkable, prompt functional improvement, even beyond the theoretical period of consolidation of an outcome after the beginning of a disease. Naturally, we unequivocally respect a certain rhythm by integrating the duration of the patient’s processing of physical/mental skills and the ensuing fatigue. Overall, the strong stimulation of the sensory, sensitive, and motor spheres promotes and interacts with the mechanisms related to the performance of the tasks in the cognitive and emotional domains, through the activation of multiple neural networks [46,47]. The degree of change associated with neuroplasticity through hippotherapy is most likely linked to both the relevance of the activity and the intensity and frequency of the elements that constitute it [48,49,50], and represents a strong imprinting for functional and behavioral remodeling. The protocol timeline was optimized considering that potential strong imprint brought during the initial session. One of the main themes that came up during the six-month period was related to fear (relapse, the future, not achieving the goals, pain, relationship issues).

*Supportive care treatment: *Participants included in both groups received therapeutic support at the MIS. This treatment was fully personalized and adapted to each participant’s needs. Indeed, breast cancer and its treatments induce symptoms and distress that have dramatic repercussions on the lives of patients and their loved ones (anxiety, sleep disorders, psychological difficulties, pain, eating disorders, weight gain or loss, hot flashes, addictions, social, personal or professional difficulties, and sexual difficulties, among others). From the onset of the disease, specialized health professionals and experts from the social sector (algologists, sexologists, addictologists, occupational physicians, homeopaths, acupuncturist, physical therapists, psychologists, nutritionists, social workers, etc.) accompanied each patient in order to prevent or alleviate these symptoms, and thus preserve the best possible quality of life. A general scheme of supportive care is presented in Table 3.

In the hippotherapy-treated group, patients interrupted their supportive care and replaced it with the hippotherapy sessions according to the project’s timeline (see Figure 1).

### 2.4. Questionnaires

Questionnaires were administered to the patients to evaluate their mental and physical status. The questionnaires were fulfilled in the week following the inclusion into the protocol (Q1), one week after the first session of hippotherapy for the treated group (and the corresponding period for controls) (Q2), and after the last hippotherapy session, six months after the beginning of the program (Q3). The time interval between the inclusion of patients, and the first, second and last evaluation was carefully controlled to allow a homogeneity between the two groups (see Figure 1 and Table 1). The recruitment was done in groups of four patients: a set of two patients from the hippotherapy group was systematically paired with a set of two patients from the control group. This way, they signed an informed consent in the same period (with less than two weeks of delay between each other) and the questionnaires were collected simultaneously for each patient of each group.

#### 2.4.1. Primary End Point

EORTC QLQ-C30 and EORTC QLQ-BR23: The primary end point of the study was the patient’s quality of life. It was assessed through EORTC QLQ-C30 (European Organization for Research and Treatment of Cancer Quality of Life questionnaire), supplemented by the breast cancer module (EORTC QLQ-BR23). EORTC QLQ is one of the most widely used specific instruments for measuring the QoL of cancer patients [36,51,52,53,54,55,56]. This questionnaire is designed to be specific for cancer and breast cancer, multidimensional in its structure, appropriate for self-administration (short and easy to complete), and applicable in a wide range of cultural contexts. The QLQ-C30 comprises 30 items categorized to assess physical, psychological and social issues, while the QLQ-BR23 contains 23 questions, assessing key factors in breast cancer survivors. The QLQ-C30 includes global health status, five functional scales, and three symptom scales. High scores of functional scales characterize healthy functioning. Similarly, a high score for global health status signifies a higher quality of life. On the other hand, high scores of symptom scales show a high level of problems. Scores for all scales and single items range from 0 to 100. The QLQ-BR23 includes four functional scales and four symptom scales. The “upset by hair loss” item was not taken into account, due to the low number of patient concerned. High scores of functional scales represent better functioning, and high scores of symptom scales indicate more issues. On average, the QoL questionnaire takes 11 to 12 min to be completed, with little or no assistance.

The summary scores are computed as follows: 

EORTC QLQ-C30 Summary Score = (Physical + Role + Social + Emotional + Cognitive + 100-Fatigue + 100-Pain + 100-Nausea_Vomiting + 100-Dyspnoea + 100-Sleeping Disturbances + 100-Appetite Loss + 100-Constipation + 100-Diarrhoea)/13.

EORTC QLQ-BR23 Summary Score = (Body image + Sexual functioning + Sexual enjoyment + Future perspectives + 100-Side effects from treatment + 100-Breast Symptoms + 100-Arm Symptoms)/7.

#### 2.4.2. Secondary End Points

FACT-Cog V3: Cancer and its treatment can lead to an impairment of cognitive functions. The FACT-Cog questionnaire aims at evaluating the effectiveness of hippotherapy to reduce these effects [57,58,59]. It is a tool designed specifically for breast cancer patients who have received chemotherapy treatment. The questions are grouped into six cognitive domains (memory, verbal fluency, concentration, mental sharpness, resistance to interference, multitasking ability) allowing to calculate four subscales (perceived cognitive abilities, perceived cognitive impairments, comments from others, and impact of perceived cognitive impairment on quality of life). It is a semi-quantitative, self-reporting 5-point Likert scale (from 0 “not at all” to 4 “very much”) where the higher the scores, the higher the cognitive perception or quality of life.

MFI-20: Among the instruments to measure fatigue status (the most common complaint of breast cancer survivors), the MFI-20 seems to be one of the most reliable questionnaires to provide a fatigue profile [60,61,62]. Its psychometric properties have been studied in different populations and it is easy to administer. It is a semi-quantitative self-reporting 5-point Likert scale (from 1 “not at all” to 5 “very much”). Each subset score (general fatigue, physical fatigue, mental fatigue, motivational fatigue, and activity reduction) ranges from 4 to 20. The global fatigue score ranges from 16 to 100. A high score corresponds to increased fatigue. 

HADS: This tool aims to identify anxious/depressive symptoms and to evaluate their severity [63,64,65]. It does not attempt to distinguish between different types of depression or anxiety states. This scale was constructed by excluding any items concerning somatic aspects, as they could lead to confusion between physical and mental illness. It is a self-reporting questionnaire to be completed according to one’s condition during the past week. There are 14 questions, 7 for each subset. Scores range from 0 to 21 in each subscale, with higher scores indicating higher levels of anxiety or depression. The scale is presented as a reliable instrument for separately screening clinically significant anxiety and depression. On average, it takes 2 to 6 min to be completed.

BIS: One aspect that is strongly impacted by breast cancer is the change of the patients’ body perception [66,67]. The study of the body image in oncology focuses more on the subjective experience of different related aspects than on the perceptual aspects. The BIS questionnaire focuses on patients’ emotional and behavioral experiences of their body image, resulting from cancer and its treatment, including aspects of perceived physical appearance, body integrity, and seduction capacity. The BIS was designed as a brief instrument (10 questions), complementary to EORTC QLQ-C30. It is a self-reported semi-quantitative 4-point Likert scale (from 0 “never” to 3 “often”). The score ranges from 0 to 30. A high score indicates an impaired body perception.

Further information on the research design is available in Appendix B.

## 3. Results

### 3.1. Both Groups Are Demographically and Clinically Equivalent 

The demographic features of the patients from the two groups were fairly homogenous, and did not differ significantly (see Table 1). In addition, their hormonal statuses showed the same distribution across the two groups (non-menopausal and menopausal). Although the types of breast cancer, according to the anatomopathological studies, were not completely homogeneous between the two groups, the stage, grade and affected side were comparable (see Table 1). Lastly, the type of cancer treatment given to the patients did not differ significantly: radiotherapy for 6½ weeks, hormonal therapy (Tamoxifen or Exemestane or Letrozole) for 5 years, and chemotherapy depending on the lesion (Epirubicin-Cyclophosphamide + Paclitaxel or Docetaxel ± Trastuzumab).

We also analyzed and compared the evaluated items obtained before the start of the care program, to see whether the two groups had the same baseline scores (Table 2). Indeed, the scores were homogenous, showing no statistical differences regarding the results of the initial questionnaires, whatever the measured function, except for one symptom item of the EORTC QLQ-C30 (the dyspnea symptom). In this regard, all patients in the trial were assessed for cardiac function. None of the patients in any group showed evidence of cardiac dysfunction. Although dyspnea values were significantly higher in the hippotherapy group at baseline, they decreased as early as one week after the start of the trial (see Appendix C; Table A1 and Table A2); therefore, the initial difference was not considered relevant. Considering the overall number of items tested (thirty-five), and the fact that only one subset in the one of the scores calculated resulted in being significantly different, we can consider that the two groups presented a homogenous baseline and were similar.

### 3.2. Quality of Life (QoL) Is Enhanced by Hippotherapy

QoL is a multidimensional concept that encompasses physical, psychological and social well-being. According to the WHO, it is defined as the individual’s perception of their position in life in the context of the culture and value systems in which they live, and in relation to their goals, expectations, standards and concerns [68]. Its improvement is one of the key objectives during and after a breast cancer treatment process.

Participants had to answer a standardized QoL questionnaire adapted to cancer patients (EORTC QLQ-C30) and a QoL questionnaire specifically designed for breast cancer patients (EORTC QLQ-BR23) [52,55,56,69]. The EORTC QLQ-C30 subset scores enabled to assess a global quality of life, labeled as “summary score”. It covered: functional scales of physical, role, emotional, cognitive and social functioning, and symptom scales of fatigue, nausea/vomiting, pain, dyspnea, insomnia, appetite loss, constipation and diarrhea, financial difficulties and an assessment of the global health status/QoL (all ranging from 0 to 100). For the summary score and the functional items, an increased score corresponded to an improvement of the corresponding QoL. For the symptoms scale, the lower was the score, the higher was the QoL. The EORTC QLQ-BR23 subset scores enabled us to assess the breast cancer patients’ QoL, also labeled as “summary score”. This score assessed the QoL of patients, targeting specifically the items which might be impacted after breast cancer therapy. It covered functional scales of body image, sexual functioning, sexual enjoyment, and future perspectives, and symptom scales of side effects from treatment, breast symptoms, and arm symptoms.

We verified whether these QLQ-C30 scores were impacted by the two different treatments (Figure 2A). This resulted in a significant change over time of the QoL summary score for the group following the hippotherapy treatment (Friedmann Chi-squared test = 27.706; df = 2; *p*-value = 9.631 × 10^−7^). Conversely, no change was observed in the control group (Friedmann Chi-squared test = 2.902; df = 2; *p*-value = 0.2343). We confirmed these results through ad hoc tests for the hippotherapy group, showing an increase of QoL after the first week of treatment (two-sided paired Wilcoxon test between first and second measurement times: V-statistics = 101, effect size = 0.60795, *p*-value = 0.0001263). This improvement was maintained after the end of the program, i.e., six months after the first week of treatment (two-sided paired Wilcoxon test between first and third questionnaire: V-statistics = 44, effect size 0.745, *p*-value = 9.399 × 10^−6^) but no statistical difference was observed between the second and third questionnaire (two-sided paired Wilcoxon test: V-statistics = 280, effect size = 0.139, *p*-value = 0.414). We then tested the differences in the evolution of the QoL score, with respect to the initial state of each patient, i.e., baseline correction was performed, as described in Methods (Figure 2B). Moreover, we assessed the difference in the evolution of these values between the two groups using a robust between–within participants ANOVA. This test showed that the evolution of the QoL score differed significantly between the two groups (F-value = 12.4347, df = 2, *p*-value = 0.0014). Then, ad hoc tests showed a significant increase in the QoL summary score in the hippotherapy group compared to the control group at the first post-treatment evaluation (two-sided unpaired Wilcoxon test: W-statistics = 687, effect size = 0.397, *p*-value = 0.001826) and at the final evaluation times (two-sided unpaired Wilcoxon test: W-statistics = 595, effect size = 0.305, *p*-value = 0.008539). 

Lastly, we tested the differences in the QoL changes for each QLQ-C30 subset scale (Figure 2C,D). Globally, the behaviors of the two groups differed significantly at the first post-treatment evaluation (robust between–within participants ANOVA F-value = 4.3862, df = 2, *p*-value = 0.0023) and final evaluation (robust between–within participants ANOVA F-value = 6.4769, df = 2, *p*-value = 0.0012) (see Appendix C; Table A3 for ad hoc tests). The changes were significantly improved in the treated group for all the functional subsets, except for the physical functioning at the first post-treatment evaluation, and social functioning at both evaluation times (Appendix C; Table A3*)*. Some symptom subsets, i.e., nausea, dyspnea, fatigue and symptoms from treatment, were also significantly improved in the treated group compared to the control group at the final evaluation time (Appendix C; Table A3). Interestingly, it has been shown that a change in the baseline score by more than 10 points indicates changes in QoL that are perceptible by the patient [70]. This threshold has been reached for the hippotherapy group for almost all the functional scales (except social and physical functioning), and for the fatigue, and dyspnea items of the symptoms scales. This needs to be highlighted as it shows that hippotherapy benefits are directly perceptible by the patients.

This analysis has been replicated for the QLQ-BR23 score (Appendix C; Figure A1). Similar results to those of the QLQ-C30 were observed for the summary score. They show an improvement across time for the treated group only (Friedmann Chi-squared test = 14.392, df = 2, *p*-value = 0.0007497 for hippotherapy group; Friedman Chi-squared = 2.6182, df = 2, *p*-value = 0.2701 for control group). The ad hoc two-sided paired Wilcoxon post-test showed that the score increased significantly between the baseline and first post-treatment evaluation (V-statistics = 96.5, effect size = 0.619, *p*-value = 0.0002091) and between the baseline and final evaluation (V-statistics = 97, effect size = 0.596, *p*-value = 0.0003674). It remained stable between first post-treatment evaluation and final evaluation (V-statistics = 310, effect size = 0.0602, *p*-value = 0.7269). The changes of the scores after baseline correction are significantly different between the hippotherapy and the control group (robust between-within participants ANOVA F-value = 12.1600, df = 2, *p*-value = 0.0014). The hippotherapy group showed a higher improvement at both post-treatment measurement times (Appendix C; Figure A1A,B—ad hoc tests with two-sided unpaired Wilcoxon test: for the first post-treatment evaluation, W-statistics = 672, effect size = 0.284, *p*-value = 0.0234; for the final evaluation, W- statistics = 641, effect size = 0.304, *p*-value = 0.01707). Similarly, for the QLQ-C30, major changes were observed for the functional subsets, where improvements were higher for the hippotherapy group than for the control group (robust between–within participants ANOVA—first post-treatment evaluation F-value = 4.1610, df = 2, *p*-value = 0.0184; final post-treatment evaluation F-value = 3.2578, df = 2, *p*-value = 0.023) (see Appendix C; Figure A1C,D and Table A4 for ad hoc tests).

### 3.3. Hippotherapy Speeds Up Cognitive Recovery

Cognitive deficits (e.g., in attention, concentration, executive function, working memory) are commonly observed after cancer treatments. These deficits are supposed to be more severe if the cancer therapy involved chemotherapy [71,72]. To quantify the effects of hippotherapy on cognitive function compared to controls, we used the FACT-Cog questionnaire [59]. This questionnaire allowed us to estimate the evolution of the patients’ cognitive abilities through four cognitive items: the perceived level of cognitive abilities (ranging from 0 to 36), the perceived level of cognitive impairment (ranging from 0 to 80), the impact of the cognitive state on the patient’s quality of life (ranging from 0 to 16), and the comments from other people on the patient’s cognitive abilities (ranging from 0 to 16). We calculated a mean of these scores to estimate a “summary cognitive score”. Doing so permitted us to quantify the global cognitive state of the patients (Figure 3). We analyzed the evolution of the FACT-Cog summary score over time (Figure 3A left panel). The score showed a favorable evolution across treatment only for the hippotherapy group (Friedmann chi-squared test = 11.881, df = 2, *p*-value = 0.002631 for hippotherapy group; Friedman chi-squared test = 4, df = 2, *p*-value = 0.1353 for control group). It was significantly higher between the baseline and first post-treatment evaluation (two-sided paired Wilcoxon test V-statistics = 32, effect size = 0.630, *p*-value = 0.00391) and between the baseline and final evaluation (two-sided paired Wilcoxon test V-statistics = 26.5, effect size = 0.613, *p*-value = 0.003574). Conversely, it remained stable between the first post-treatment evaluation and final evaluation (two-sided paired Wilcoxon test V-statistics = 111.5, effect size = 0.0173, *p*-value = 0.9031). Moreover, after baseline correction (Figure 3B), the changes observed at the first post-treatment evaluation and final evaluation were significantly different between the two groups (robust between–within participants ANOVA F-value = 8.7016, df = 2, *p*-value = 0.0066). The scores showed a higher improvement for the hippotherapy group compared to the control group at the first post-treatment evaluation time (two-sided unpaired Wilcoxon test W-statistics = 546, effect size = 0.441, *p*-value = 0.00123) and at the final evaluation time (two-sided unpaired Wilcoxon test W-statistics = 281.5, effect size = 0.296, *p*-value = 0.02425).

Lastly, we tested the differences in the cognitive deficits for each subset scale (Figure 3C,D). Globally, the behaviors of the two groups were significantly different at the first post-treatment evaluation (robust between–within participants ANOVA F-value = 12.219, df = 2, *p*-value = 0.0019) and final post-treatment evaluation (robust between–within participants ANOVA F-value = 6.6277, df = 2, *p*-value = 0.0150) (see Appendix C; Table A5 for ad hoc tests). Moreover, two of the four items, the perceived cognitive impairment and the impact of cognitive abilities on quality-of-life, showed a higher improvement at the first post-treatment evaluation and final post-treatment evaluation compared to baseline in the hippotherapy-treated group.

### 3.4. Hippotherapy Reduces Fatigue, Anxiety, and Depression

Fatigue is a persistent feeling of emotional, cognitive, and physical exhaustion in patients diagnosed and treated for breast cancer. It is the most common symptom experienced by breast cancer survivors, with prevalence rates ranging from 15–99%, and is unrelated to recent activity and unrelieved by rest [5,6]. It is also currently associated with anxiety and depression. To explore the effect of hippotherapy on fatigue symptoms after breast cancer, we used the MFI-20 questionnaire [62,73]. This questionnaire allowed to calculate several fatigue scores allocated to five subsets: general fatigue, mental fatigue, physical fatigue, motivational fatigue, and activity reduction. We also calculated an overall score based on the sum of all subsets. 

The summary score showed a significant improvement across time in the treated group but not in the control group (Figure 4A; Friedman Chi-squared = 16.623, df = 2, *p*-value = 0.0002457 for hippotherapy; Friedman Chi-squared = 0.58491, df = 2, *p*-value = 0.7464 for controls). These improvements were significant between the baseline and first post-treatment evaluation (two-sided paired Wilcoxon V-statistics = 508, effect size = 0.535, *p*-value = 0.001608) and between the baseline and final evaluation (two-sided paired Wilcoxon V-statistics = 546.5, effect size = 0.715, *p*-value = 2.138 × 10^−5^). Between first post-treatment evaluation and final evaluation, the changes observed were not significant (two-sided paired Wilcoxon V-statistics = 393, effect size = 0.270, *p*-value = 0.1042). Furthermore, after baseline correction, the changes observed at the first post-treatment evaluation and final post-treatment evaluation were significantly different between the two groups (Figure 4B; robust between–within participants ANOVA F-value = 8.4327 df = 2, *p*-value = 0.0039). The scores showed a larger improvement for the hippotherapy group compared to the control group at the first post-treatment evaluation (two-sided unpaired Wilcoxon test W-statistics = 334.5, effect size = 0.271, *p*-value = 0.03185) and final evaluation (two-sided unpaired Wilcoxon test W-statistics = 274, effect size = 0.358, *p*-value = 0.003591). 

Lastly, we explored the evolution after baseline correction of the scores obtained for each different fatigue subset. Globally, the behaviors of the two groups were significantly different at first post-treatment evaluation (robust between-within participants ANOVA F-value = 6.8586, df = 2, *p*-value = 0.0124) and final post-treatment evaluation (robust between-within participants ANOVA F-value = 6.2769, df = 2, *p*-value = 0.0164) (see Appendix C; Table A6 for ad hoc tests). At the first post-treatment evaluation, general and mental fatigue showed a significant improvement in scores for the hippotherapy group compared to the control group (Figure 4C and Appendix C Table A6 for statistics). At the final evaluation, almost all the subsets of the fatigue questionnaire showed a significant improvement in the hippotherapy group (Figure 4D and Appendix C Table A6 for statistics).

Symptoms of depression and anxiety affect up to a quarter of breast cancer survivors. Survivors have a 60% increased risk of developing depression, anxiety and stress-related disorders within 10 years of cancer diagnosis compared to the general female population [6,74]. In more than half of survivors, fear of recurrence is present and may increase the risk of developing mental health problems. In addition, depression has been associated with a higher risk of cancer recurrence [75].

To explore the effect of hippotherapy on anxiety and depression, we analyzed the evolution of the scores obtained from the HADS questionnaire [63,76,77]. This questionnaire was divided into two subsets, the anxiety subset, and the depression subset, each containing seven items. We observed that anxiety and depression scores decreased significantly over time in the hippotherapy group (Figure 5A,C; Friedmann Chi-squared test = 27.421, df = 2, *p*-value = 1.11 × 10^−6^ for anxiety; Friedmann Chi-squared test = 13.339, df = 2, *p*-value = 0.001269 for depression). Regarding the score of anxiety, a significant improvement was noticed between the baseline and first post-treatment evaluation (two-sided paired Wilcoxon test V-statistics = 324.5, effect size = 0.547, *p*-value = 0.001088), between the baseline and final evaluation (two-sided paired Wilcoxon test V-statistics = 462, effect size = 0.832, *p*-value = 2.295 × 10^−6^) and between the first post-treatment evaluation and final evaluation (two-sided paired Wilcoxon V-statistics = 355.5, effect size = 0.412, *p*-value = 0.01135), revealing a continuous progression across time. Concerning the depression score, a significant improvement was shown between the baseline and first post-treatment evaluation (two-sided paired Wilcoxon test V-statistics = 336.5, effect size = 0.441, *p*-value = 0.009611), between the baseline and final evaluation (two-sided paired Wilcoxon test V-statistics = 302, effect size = 0.556, *p*-value = 0.001332), and between the first post-treatment evaluation and final evaluation (two-sided paired Wilcoxon test V-statistics = 353.5, effect size = 0.356, *p*-value = 0.03791), also showing a sustained progression across time in the treated group. 

Regarding the control group, only anxiety decreased significantly over time (Friedmann Chi-squared test = 8.0638, df = 2, *p*-value = 0.01774 for anxiety; Friedman Chi-squared test = 0.58333, df = 2, *p*-value = 0.747 for depression). Interestingly, the decrease was not significant between the baseline and first post-treatment evaluation (two paired Wilcoxon V-statistics = 99, effect size = 0.174, *p*-value = 0.29), but an improvement was noticed between the baseline and final evaluation (two-sided paired Wilcoxon V-statistics = 261, effect size = 0.584, *p*-value = 0.001499) and between the first post-treatment evaluation and final evaluation (two-sided paired Wilcoxon V-statistics = 283, effect size = 0.509, *p*-value = 0.006199; Figure 5A,C). 

We finally explored the differences observed after baseline correction between the two groups at the first post-treatment evaluation and final post-treatment evaluation. The analysis revealed that improvements from baseline score were larger in the hippotherapy group compared to the control group for both anxiety and depression (robust between–within participants ANOVA F-value = 4.6659, df = 2, *p*-value = 0.0309 for anxiety; robust between–within participants ANOVA F-value = 5.6525, df = 2, *p*-value = 0.0060 for depression). These changes were significantly higher at the first post-treatment evaluation (two-sided unpaired Wilcoxon W-statistics = 314.5, effect size = 0.280, *p*-value = 0.02569 for anxiety; two-sided unpaired Wilcoxon W-statistics = 338.5, effect size = 0.283, *p*-value = 0.0239 for depression) and at the final evaluation (two-sided unpaired Wilcoxon W-statistics = 355, effect size = 0.195, *p*-value = 0.0419 for anxiety; two-sided unpaired Wilcoxon W-statistics = 316.5, effect size = 0.267, *p*-value = 0.03755 for depression; Figure 5B,D).

### 3.5. Hippotherapy Improves Body Image

Body image is a crucial endpoint in the field of breast cancer. Indeed, body image is subjected to important changes due to diagnosis and treatment. Each method of treatment (surgery, chemotherapy, radiotherapy, and hormone therapy) or their combination may result in major alterations in a patient’s appearance, or have a undeniably negative impact on the way the treated patient perceives their body [78,79,80]. We compared the evolution of the body image scale [66] over time for the hippotherapy and control group (Figure 6).

A significant improvement of body perception was found in the group treated by hippotherapy (Friedmann Chi-squared test = 22.797, df = 2, *p*-value = 1.121 × 10^−5^) but not in the control group (Friedmann Chi-squared test = 0.65116, df = 2, *p*-value = 0.7221). Moreover, post-hoc tests showed a continuous improvement from the baseline to final post-treatment evaluation (two-sided paired Wilcoxon V-statistics = 379, effect size = 0.696, *p*-value = 6.341 × 10^−5^ between baseline and first post-treatment evaluation; two-sided paired Wilcoxon V-statistics = 521.5, effect size = 0.715, *p*-value = 1.688 × 10^−5^ between baseline and final evaluation; V-statistics = 400, effect size = 0.349, *p*-value = 0.03306 between first post-treatment evaluation and final evaluation) (Figure 6A). We finally compared the differences observed after baseline correction between the two groups at first post-treatment evaluation and final evaluation (robust between–within participants ANOVA F-value = 12.1719, df = 2, *p*-value = 0.0060). Body image improvements were significantly higher for the hippotherapy group at the first post-treatment evaluation (two-sided unpaired Wilcoxon W-statistics = 268, effect size = 0.366, *p*-value = 0.00405) and at the final evaluation (two-sided unpaired Wilcoxon W-statistics = 269, effect size = 0.333, *p*-value = 0.0107) (Figure 6B).

## 4. Discussion

We report the evaluation of a novel framework via hippotherapy aiming at becoming the key initial support care step of integrative oncology in breast cancer [81,82] and beyond. The reported method considers the post-treated breast cancer woman in her wholeness, combining, simultaneously, the different targeted individual spheres and the health disciplines that go with them. We applied the hippotherapy approach to a cohort of patients undergoing a close and personalized follow-up for high-continuity free-access support care, and compared the evolution of this cohort with that of a control, hippotherapy-free support care cohort. Through these cohorts, we were able to examine thirty-five items covered by six acknowledged clinical scales assessing different functions related to the patients’ everyday life and their individual self-perception over time.

Despite crucial advances in the understanding of breast cancer biology, diagnosis, and treatment, several important clinical questions remain unsolved both for treatment and supportive care. The former concerns prevention, diagnosis, tumor progression, treatment, therapeutic resistance and metastasis, confirming that breast cancer is not a uniform disease [83,84]. In this context, modern systems’ biology, based on multi-omics approaches, has made a major contribution to overcome those issues in the biological domain. Indeed, multi-omics strategies and their integration in the field of breast cancer are new tools to determine the molecular signature in each case, with high potential in clinical practice [85]. Molecular profiling allows cancer cell identification at several levels: genome, transcriptome, proteome and metabolome [86]. The results of these approaches in systems’ biology, and the consequent diagnostic and therapeutic solutions, allow the medical community to foresee a cure in the near future.

Beyond the purely biological systemics, another systemic dimension is fundamental and must have a sustainable basis: the medium- and long-term life course of these patients. This combines the body, mind and spirit spheres and the repercussions on these spheres of an illness involving such symbolic violence, as well as such aggressive treatments that leave numerous deep traces in the organism and the psyche [87]. A space must therefore be solidly created to support the global reconstruction of these patients, as well as their complete integration. The prognosis and the survival rate of women with breast cancer have improved considerably worldwide [4]. Multidimensional programs for breast cancer survivors have become increasingly important in supportive care to maximize women’s quality of life for a successful transition to a normal life [81]. Indeed, because of patients seeking to enhance their well-being, improve their quality of life, and alleviate the symptoms of the disease and the side effects of current cutting-edge treatments, there is a growing body of research supporting the development of integrative therapies, particularly mind–body therapies, as effective supportive care strategies during and after breast cancer treatment [81,88]. However, many integrative practices remain understudied, and there is insufficient evidence for clinicians to either recommend or definitively avoid them. In November 2014, the Society for Integrative Oncology published clinical practice guidelines to inform both clinicians and patients about the use of integrative therapies during breast cancer treatment including managing treatment-related symptoms [88]. These 2014 clinical practice guidelines were derived from a systematic review of randomized clinical trials published between 1990 and 2013. However, in the context of integrative oncology, it was stated that these recommendations only signified that a specific support therapy should be considered as a viable, but not the sole, option for managing a specific symptom or side effect [81].

Support care is a multilayered intervention in which the active constituents are not always accurately identifiable with respect to the needs [81,89]. While after diagnosis and treatment of breast cancer, patients exhibit patent clinical heterogeneity and confounding factors, clinicians may have real difficulty incorporating the patient/caregiver perspective into decision-making, and many issues essential to the patient may be overlooked. Care in these circumstances is dictated by the homeostatic, laws where survival is the main issue and going beyond is physically and psychically painful and experienced as unattainable. In this context, it is essential to evoke the notion of discontinuity, of fragmentation of states of consciousness [45]. The period between diagnosis, treatment and supportive care until the woman’s reintegration, or not, into a semblance of normal life deprives her of a more or less important part of her reference points, and requires her to maintain a semblance of continuity. To make this possible, the traumatic experiences and the associated states of consciousness become sequestered “islets”, separated and invisible from the other states accessible to consciousness at a given moment. The woman, in an instinct for survival, will put aside or transform the memory of the event that she unconsciously considers as endangering her, weakening her. It is in this dissociated terrain that the person tries to maintain a foothold, to the detriment of the solutions offered to her.

Within this framework, as long as the woman is not in possession of her means, her life and her psycho-corporal unity, she will be unable to choose and integrate supportive care, because her choice will not necessarily be relevant. Indeed, in this period of extreme vulnerability, it is unlikely that the choice will be made in an area that might push the patient out of her comfort zone. She will certainly not have the capacity to “put herself in danger again”, or to make a choice that requires her to make an extra effort. In order to lead her towards introspection and in-depth reconstruction, it is essential to provide her with a very specific and containing context. Hippotherapy provides this context in several ways. First, there is the powerful symbolism of the horse carrying and moving forward. Then, there is the fundamental notion of the patients being set into motion. During the period of diagnosis, treatment and convalescence, time stands still and the individual with it, so as not to endanger this precarious homeostasis. Hippotherapy allows for this setting into motion, which is both physiological, because it is biomechanically similar to human movement during walking [24], global, because it simultaneously engages the physical, cognitive, emotional and social spheres [8,9,10,11,12,13,14], and which offers the person a non-restrictive elsewhere, far from the clinical environment (the patient progressively moves from the context of grief to the discovery of a large palette of personal potentials). 

Being set into motion in hippotherapy acts directly on the body and contextualizes it. The septo-hippocampal system (which includes the hippocampus, lateral septum, and medial septum, and communicates with other key central nervous system (CNS) structures, such as the amygdala and the hypothalamus) has been shown to be involved in a number of key functions, such as movement and context coding, as well as in emotional and motivational behaviors [90,91,92]. The colocalization of these different functions and their possible modulation in these structures have fundamental functional implications here. It is thus likely that the alterations in emotional regulation (fear, rage, anxiety, depression, demotivation, sexual dysfunction) present in breast cancer patients and reinforced by treatments are likely to disrupt, in turn, many everyday situations requiring a context–action association [92]. Instead, the lateral septum can transmit a reward signal proportional to movement through changes in the level of activation of place (context), movement and reward-related circuits. Putting the patient back into motion in a nonrestrictive context that strongly stimulates both reward and motivation is therefore a fundamental lever in the up-regulation of the remodeling of brain activity, underlying the different functions, symptoms and mental representations [92].

Through contextualized and relevant multimodal inputs (sensory, exteroceptive, proprioceptive, interoceptive and emotional), hippotherapy has a robust action on the motor abilities of the person [10,12,14,93]. In parallel to hippotherapy’s setting into motion, the work of understanding the instantaneous needs of the patients is fundamental. This was conducted within a framework of decontextualization in relation to the patients’ pathology, which reduces the stigma and, consequently, the level of anxiety of breast cancer survivors. This framework included two levels: the therapist listening, which allows him/her to constantly adapt the exercises, and the patient’s listening to their own body and its possibilities, without any necessary cognitive elaboration on their part. Still here, the hippocampus will play a major integrating role by participating in the processing of sensory inputs, according to the context [94]. Indeed, to adapt their behavior in their daily lives, individuals must first detect and respond to changes in external and internal environments. Sensory and motor systems exhibit remarkable responsiveness and plasticity in their structures and functions in order to adapt to constantly changing environmental conditions [95,96]. Sensorimotor processing is overly complex, since it is strongly influenced by learning, perception, motivation, context, and the state of the brain (being also affected by many pathologies) [94]. The hippocampus has been shown to respond to somatosensory [97], visual [98], auditory [99], and olfactory [100] stimuli and has been considered as a top-level sensory area [101]. It automatically anticipates and synthesizes representations of the world far beyond the real-time inputs of the sensorium [102]. Hippotherapy, by stimulating the hippocampus, might be the basis for the remarkable and early reinforcement of role functioning and social functioning, enhanced integration of body image, and an improved representation of future perspectives in these patients. The positive effects on physical aspects would also be supported by the integration of multimodal inputs at the level of the hippocampus–lateral septum–amygdala–hypothalamus–ventral tegmental area–prefrontal cortex system. This integration would have a direct relieving impact on fatigue and pain, as well as a reversal effect on usually observed activity reduction, as shown in our data when compared to the control group.

From the point of view of changes at the molecular level that support functional improvement, some aspects are essential to consider. The lessons learned to date from studies on Parkinson’s disease [103] and on several psychiatric diseases [104] regarding alterations in gut physiology and degenerative changes in the CNS can be reasonably applied to the field of brain plasticity through hippotherapy by a reverse reasoning. Indeed, a convergence of multidisciplinary studies highlights the complexity of the bidirectional signaling pathways between the gut and the brain. Among others, signaling depends on substantial vagal efferent nerve transmission from the enteric nervous system (ENS) through reciprocal sensory vagal stimulation of brainstem nuclei that project to rostral CNS areas involved in cognition and complex behaviors [105]. Indeed, ascending vagal afferents from the gut to the brain account for 80–90% of the 80,000 to 100,000 vagal nerve fibers. This signaling relies also on hormonal stimulation through the hypothalamic–pituitary–adrenal axis. The ENS produces more than 30 neurotransmitters (hormones and peptides) that are released into the bloodstream. They have the ability to cross the blood–brain barrier and can act synergistically with the vagus nerve [104] in the CNS. The disruption of bidirectional signaling between the gut and the brain appears to be closely related to the etiology of affective spectrum disorders and other psychiatric diseases [104,106,107].

It has been shown that self-induced movement has key effects on the synthesis and release of hormones and peptides strongly involved in brain modulation [108,109,110]. Exercise has been suggested as the ‘must’ for the improvement of quality of life after breast cancer [111,112,113,114]. However, patients are constantly dealing with fatigue, sleeping disorders and pain, which may prevent them from regular voluntary exercise routines. Consequently, engaging breast cancer survivors in physical activity is currently challenging for health care professionals [115]. Hippotherapy initiates a physiologic passive movement that rapidly becomes self-induced. This close-to-human gait movement is strongly imprinted in the patient’s body as their own locomotion and rhythm [24]. Within this movement framework, the patient doesn’t feel that he is putting himself in danger and has the space to receive rewards (experiencing pleasure, relaxation, and feeling ready to let go). Hippotherapy, therefore, might have a therapeutic effect on the regulation of bidirectional signaling between the ENS and CNS, involving the endogenous pharmacotherapy path [116]. Vagus nerve stimulation has been shown to exert a significant effect on arousal and the brain activation state through a direct effect on the concentration of key neuromodulatory substances (serotonin, dopamine, melatonin, norepinephrine, acetylcholine, GABA) at the brain level [112,117,118]. Improved regulation of mood, motivation, and reward due to the direct effect of hippotherapy on the ENS (and CNS) may be the basis for rapid improvement in functions such as emotional regulation, stamina, appetite, sexual functioning, physical and mental efficiency (concentration, mental acuity, verbal fluency, executive functions, decision making). In addition, self-perceptions such as emotional, cognitive and social functioning, notions of pleasure, or relief of fatigue (general, mental, physical, motivational) might be positively impacted. Furthermore, other neuromodulatory substances probably play a major role in the restoration of a homeostasis distinct from that of the usual survival mode of these patients. Indeed, the dialogue between the ENS and the CNS through the stimulation provided in the context of hippotherapy is likely to promote the up/down-regulation of the synthesis and release of other neuroactive substances: brain-derived neurotrophic factor (BDNF), involved in the consolidation phenomena of structural plasticity and behavioral adaptation [29,119,120]; neuropeptide Y (NPY), which controls inflammatory processes, pain, emotions, mood, cognition, stress resistance, and energy homeostasis [121,122]; vasoactive intestinal peptide (VIP) which regulates hippocampal memory processes, the central biological clock, sleep, and inflammation [123,124,125]; and plausibly, endogenous opioids and endocannabinoids in the control of pain, inflammation and stress [126,127,128]. Finally, the decrease in the levels of some key hormones related to stress and the immune response has been suggested to occur after hippotherapy [129]. Thus, this modulation at the molecular level might be the chemical substrate for the early improvement of symptoms such as pain, insomnia, anxiety, depression and other side effects, as well as for the sustainable therapeutic effect demonstrated in the hippotherapy group.

Finally, the interaction between the patient and the horse strengthens the stated achievements. Indeed, far beyond the hierarchical rapport between the clinician and their patient, imposed by the serious nature of the situation during this illness and by rooted predefined societal roles [130,131,132], the interaction between the horse and the patient constitutes a powerful basis of appeasement and self-assurance through mutual collaboration. The horse scans the patient at each encounter through its highly refined perceptive abilities, without any value judgment or cognitive elaboration that could confuse and impact their different spheres (emotional, physical, cognitive, spiritual) [133]. The woman can focus on her self-perception and well-being without the pressure and distress of an unequal interaction. After the initial encounter with the horse and the waiving of any potential apprehension, she feels trust grow in her, which helps her to open up and let go of her true nature. An in-depth work can start to achieve substantial changes in all spheres.

The different aspects assessed regarding the functioning of the patient’s body, mental and emotional dimensions in this study clearly showed a very significant improvement quite soon after the beginning of hippotherapy (i.e., after the first week of the care), compared to the group that followed a conventional supportive care pathway only. This improvement evolved and remained significantly larger than that of the control group until the end of the protocol. Some basic ideas were conveyed in the hippotherapy approach: the notions of systemics and synergy of the different members of the therapeutic team in real-time with a permanent adaptation to the patient’s output; the durability of the changes initiated after the patient’s setting into motion and going with the flow with a great availability, capacity of listening and self-adjustment; the notion of the ripple effect and its impact, where the actions initiated will provoke changes beyond the immediately noticeable or conceivable; the prioritization of the self, based on one’s own rhythm and choices. The action in a decontextualized panorama of the disease allowed each woman to work for self-renewal of continuity and unity of her states of consciousness. She was able to work out very quickly in relation to the fear of death and the choice before her, not to take everything but to be more selective about what is important by listening to her intuition, and by reappropriating her body after having abandoned it to medicine. This reawakening of her vital force and of her intrinsic motivation create the ideal conditions for an emotional liberation and a reinforcement of the unity of the being (see Table 4). With this in hand, the woman will be able to take full advantage of the already available integrative oncology supportive care path. She will be more selective and self-confident in her choices. In this way, the gap between healing and recovery is perfectly bridged by hippotherapy.

## 5. Conclusions

This work provides a paramount, clinically relevant resource of a time-efficient model for supportive care intervention after breast cancer. We emphasize a search for continuity and unification of each sphere of the patient, for which advances in global health management are urgently needed. We show that this model is conducive to the realization of a functional precision medicine, in real-time, parallel to primary clinical care, allowing to start the support care phase in a very efficient way. The results are convincing, whatever the item evaluated, among the thirty-five items chosen to characterize the patient’s global state from a physical, mental, emotional and social point of view. The positive impacts were noticed very quickly after the beginning of the treatment when comparing the hippotherapy group to the control group.

The use of integrative support care therapies during and after breast cancer diagnosis and treatment in the context of integrative oncology has been highly recommended [88]. However, these recommendations consider each support therapy as a relevant, but not the sole option, for managing a specific issue [81]. The choice is mainly given to the patient in a moment when she/he has not the personal tools to invest a deep self-work. Hippotherapy appears as the backbone and initial compulsory step of the supportive care strategy after diagnosis and primary treatment of breast cancer, preparing the patients for their self-empowerment and optimal reconstruction, and relying on the different options available to them.

“*Defects, disorders, diseases, in this sense, can play a paradoxical role, by bringing out latent powers, developments, evolutions, forms of life, that might never be seen, or even be imaginable, in their absence*”[134]

## 6. Study Limitations and Perspectives

Despite the overall convincing results of this study, and in order to give a more comprehensive view of it, some issues and flaws should be mentioned.

A possible bias exists in the fact that such protocols in the rehabilitation field cannot be designed in a double-blind manner. The only way to mitigate this issue is to carry out the evaluations blindly. Nevertheless, in our case, the evaluations were done through self-reporting questionnaires.

Since the clinical trial partly overlapped with the SARS-CoV-2 pandemic, it is difficult to evaluate the effect of the confounding factors brought by this exceptional situation on the quality of life of the patients. However, we can consider that both groups were similarly affected by the effects of the pandemic. Furthermore, the baseline scores were statistically the same.

Another issue concerns the personalized care of each patient. Unlike with clinical trials that evaluate the effect of a drug and standardize the dosage (active substance versus placebo), supportive care is individually adapted, leading to the potential heterogeneity of the groups. However, standardization of treatments in the field of supportive care or rehabilitation care is not applicable.

Even though the placebo effect is a reality in cancer clinical trials, both positive and negative effects have been described. Around a third of the patients of a controlled trial can show positive placebo effects. On the other hand, nearly a quarter of patients taking a placebo experience side effects (nocebo effects) [135]. In the context of the trial, hippotherapy is definitely not associated with being a luxurious activity, since the care is carried out in a rigorous medicalized environment of a specialized hippotherapy center, exclusively implemented at the opposite end of an equestrian luxurious environment where social and entertainment standards are given to the users.

Supplementary clinical trials involving larger cohorts will allow researchers to analyze, in depth, the influence of the patient’s profile with respect to the treatment load on the efficiency of the supportive care, as it has been shown that the more heavy the treatments (chemotherapy), the more severe the sequels are on the patient. Moreover, since the quality of life is a predictor of survival in breast cancer patients, it might be relevant to carry out a long-term follow-up of patients benefiting from hippotherapy, and measure the rate of recurrence and survival compared to conventional support care.

Inferences on CNS and ENS mechanisms through hippotherapy do not rely on direct measurements but on assumptions based on current knowledge and cross-interpretations of existing literature. The next steps should take into consideration the measurement of, for example, electroencephalography activity (EEG), functional magnetic resonance imaging changes (fMRI), functional near-infrared spectroscopy (fNIRS) changes, and the measurement of the levels of some neuromodulator metabolites (in saliva or sweat samples).

## Figures and Tables

**Figure 1 cancers-15-01317-f001:**
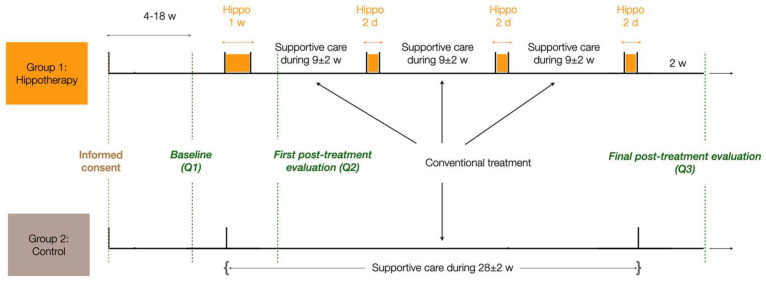
Timeline of the clinical trial: conventional support care was provided to the control group over 28 weeks, whereas the hippotherapy group received conventional supportive care for the same time, minus the days where they received the hippotherapy treatment. The aim of such a design was to prevent an unequal amount of care among groups.

**Figure 2 cancers-15-01317-f002:**
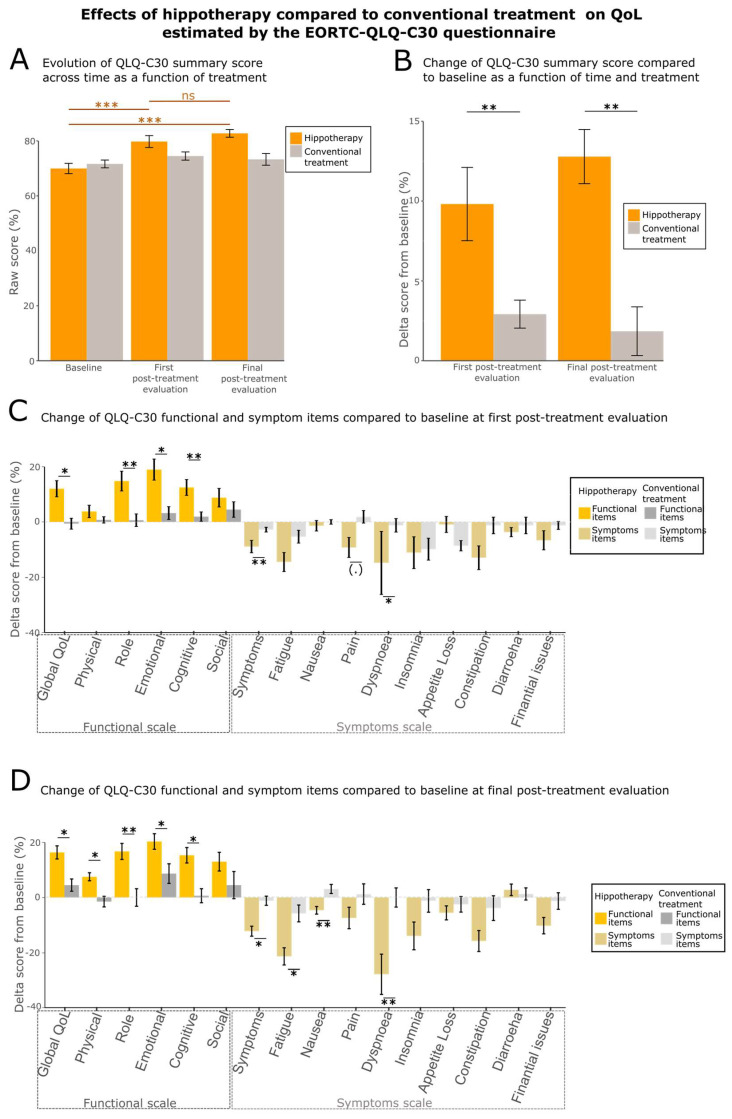
Effects of hippotherapy treatment compared to conventional treatment on QoL, estimated by the EORTC-QLQ-C30 questionnaire. (**A**) Bar chart represents means (M) and standard errors of means (SEM) of the evolution of QLQ-C30 scores across time, as a function of treatment type. Values (M ± SD) are as follows: Baseline = 70.0 ± 15.0, First post-treatment evaluation = 79.8 ± 17.3, Final post-treatment evaluation = 82.80 ± 11.2 for the hippotherapy group; Baseline = 71.6 ± 11.3, First post-treatment evaluation = 74.5 ± 11.9, Final post-treatment evaluation = 73.3 ± 17.1 for the control group. (**B**) Change in score compared to baseline QLQ-C30 score (M ± SEM), as a function of time and treatment. Values (M ± SD) are as follows: First post-treatment evaluation = 9.8 ± 18.4, Final post-treatment evaluation = 12.1 ± 13.2 for the hippotherapy group; First post-treatment evaluation = 2.92 ± 7.01, Final post-treatment evaluation = 1.89 ± 12.4 for the control group. (**C**) Comparison of score changes for the QLQ-C30 subsets between the two groups at the first post-treatment evaluation; (**D**) Comparison of score changes for the QLQ-C30 subsets between the two groups at the final post-treatment evaluation. (***) *p* < 0.001; (**) *p* < 0.01; (*) *p* < 0.05; (.) close to significance (between 0.08 and 0.05); (ns) non-significant.

**Figure 3 cancers-15-01317-f003:**
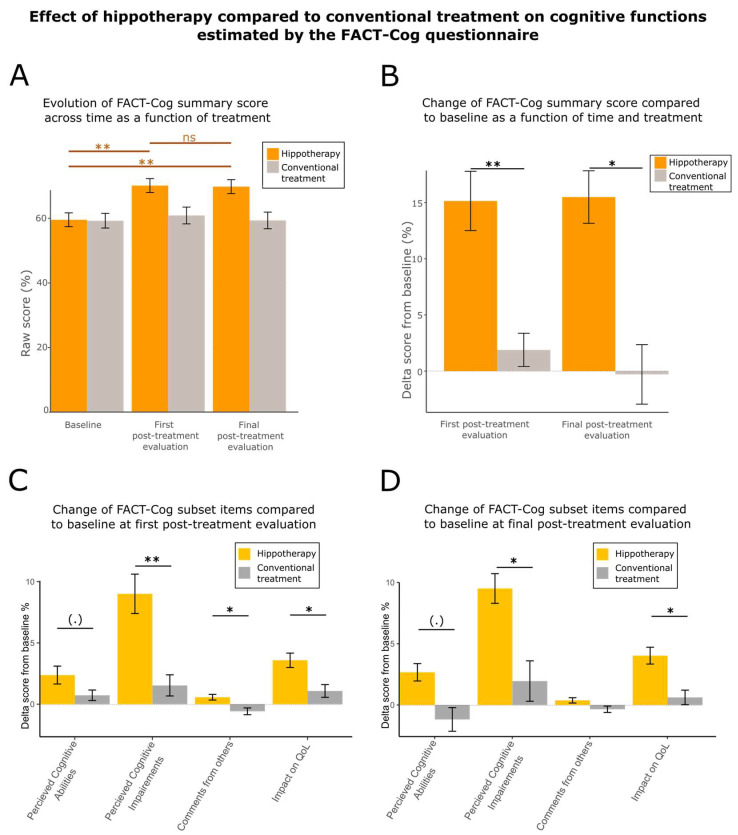
Effects of hippotherapy treatment compared to conventional treatment on cognitive functions, as estimated by the FACT-Cog questionnaire. (**A**) Bar chart represents means (M) and standard errors of means (SEM) for the evolution of FACT-Cog summary scores across time as a function of treatment’s type. Values (M ± SD) are as follows: Baseline = 89.3 ± 25.7, First post-treatment evaluation = 105.0 ± 25.9, Final post-treatment evaluation = 105.0 ± 25.8 for the hippotherapy group; Baseline = 88.9 ± 27.4, First post-treatment evaluation = 91.3 ± 31.2, Final post-treatment evaluation = 89.0 ± 30.9 for the control group. (**B**) Change of score compared to FACT-Cog baseline score (M ± SEM), as a function of time and treatment. Values (M ± SD) are as follows: First post-treatment evaluation = 15.1 ± 21.1, Final post-treatment evaluation = 14.2 ± 18.8 for the hippotherapy group; First post-treatment evaluation = 1.89 ± 11.8, Final post-treatment evaluation = 0.582 ± 21.4 for the control group. (**C**) Comparison of score changes for the FACT-Cog subsets between the two groups at the first post-treatment evaluation. (**D**) Comparison of score changes for the FACT-Cog subsets between the two groups at the final post-treatment evaluation. (**) *p* < 0.01; (*) *p* < 0.05; (.) close to significance (between 0.08 and 0.05); (ns) non-significant.

**Figure 4 cancers-15-01317-f004:**
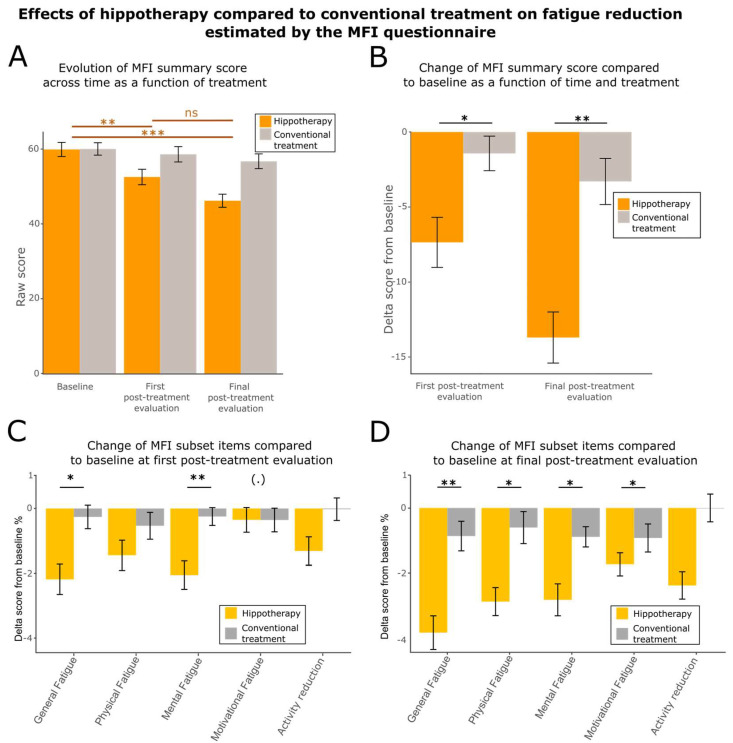
Effects of hippotherapy treatment compared to conventional treatment on fatigue reduction, as estimated by the MFI-20 questionnaire. (**A**) Bar chart represents means (M) and standard errors of means (SEM) of the evolution of MFI-20 summary scores across time, as a function of the treatment type. Values (M ± SD) are as follows: Baseline = 59.9 ± 15.1, First post-treatment evaluation = 52.5 ± 16.5, Final post-treatment evaluation = 46.2 ± 14.0 for the hippotherapy group; Baseline = 60.0 ± 13.3, First post-treatment evaluation = 58.6 ± 16.4, Final post-treatment evaluation = 56.8 ± 15.7 for the control group. (**B**) Change of score compared to MFI-20 baseline score (M ± SEM), as a function of time and treatment. Values (M ± SD) are as follows: First post-treatment evaluation = −7.36 ± 13.3, Final post-treatment evaluation = −13.3 ± 13.6 for the hippotherapy group; First post-treatment evaluation = −1.43 ± 9.22, Final post-treatment evaluation = −3.23 ± 12.5 for the control group. (**C**) Comparison of score changes for the MFI-20 subsets between the two groups at the first post-treatment evaluation. (**D**) Comparison of score changes for the MFI-20 subsets between the two groups at the final post-treatment evaluation. (***) *p* < 0.001; (**) *p* < 0.01; (*) *p* < 0.05; (.) close to significance (between 0.08 and 0.05); (ns) non-significant.

**Figure 5 cancers-15-01317-f005:**
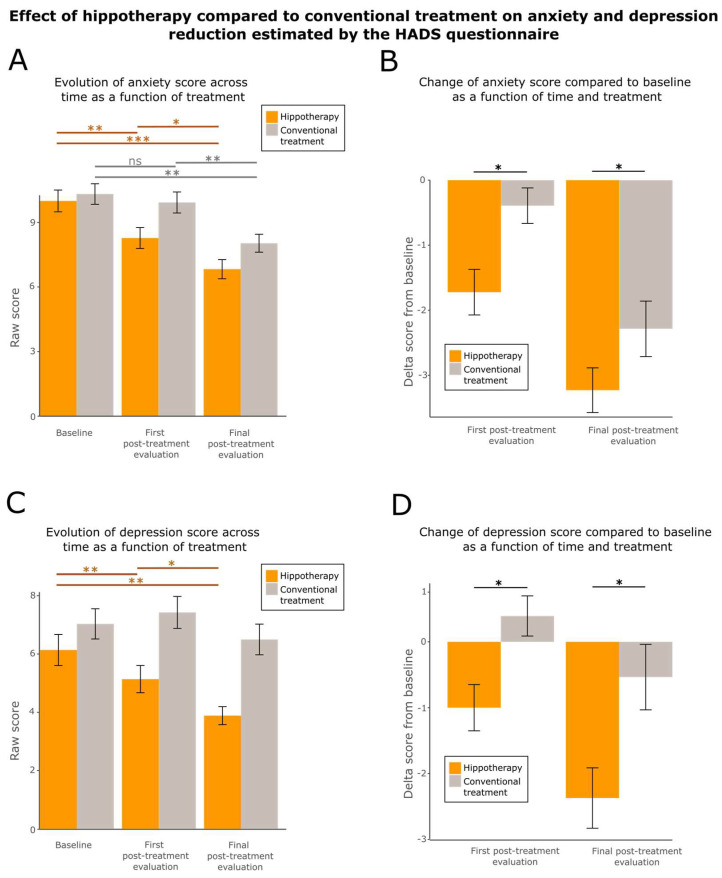
Effects of hippotherapy treatment compared to conventional treatment on anxiety and depression, as estimated by the HADS questionnaire. (**A**) Bar chart represents means (M) and standard errors of means (SEM) of the evolution of anxiety score across time as a function of the treatment type. Values (M ± SD) are as follows: Baseline = 10.0 ± 4.05, First post-treatment evaluation = 8.28 ± 3.87, Final post-treatment evaluation = 6.83 ± 3.56 for the hippotherapy group; Baseline = 10.3 ± 3.82, First post-treatment evaluation = 9.93 ± 3.91, Final post-treatment evaluation = 8.04 ± 3.33 for the control group. (**B**) Change of anxiety score compared to baseline score (M ± SEM), as a function of time and treatment. Values (M ± SD) are as follows: First post-treatment evaluation = −1.72 ± 2.80, Final post-treatment evaluation = −3.21 ± 2.78 for the hippotherapy group; First post-treatment evaluation = −0.393 ± 2.18, Final post-treatment evaluation = −2.30 ± 3.47 for the control group. (**C**) Bar chart represents means (M) and standard errors of means (SEM) for the evolution of the depression score across time as a function of the treatment type. Values (M ± SD) are as follows: Baseline = 6.14 ± 4.28, First post-treatment evaluation = 5.14 ± 3.74, Final post-treatment evaluation = 3.89 ± 2.47 for the hippotherapy group; Baseline = 7.04 ± 4.15, First post-treatment evaluation = 7.43 ± 0.39, Final post-treatment evaluation = 6.5 ± 4.21 for the control group. (**D**) Change of depression score compared to baseline score (M ± SEM), as a function of time and treatment. Values (M ± SD) are as follows: First post-treatment evaluation = −1.0 ± 2.81, Final post-treatment evaluation = −2.29 ± 3.70 for the hippotherapy group; First post-treatment evaluation = 0.393 ± 2.44, Final post-treatment evaluation = −0.481 ± 4.44 for the control group. (***) *p* < 0.001; (**) *p* < 0.01; (*) *p* < 0.05; (ns) non-significant.

**Figure 6 cancers-15-01317-f006:**
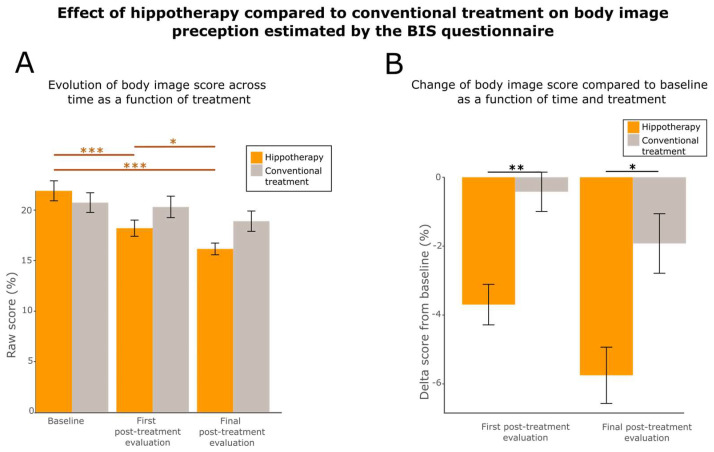
Effects of hippotherapy treatment compared to conventional treatment on body image perception, as estimated by the BIS questionnaire. (**A**) Bar chart represents means (M) and standard errors of means (SEM) for the evolution of body image score across time as a function of treatment’s type. Values (M ± SD) are as follows: Baseline = 21.9 ± 7.9, First post-treatment evaluation = 18.2 ± 6.41, Final post-treatment evaluation = 16.1 ± 4.65 for the hippotherapy group; Baseline = 20.7 ± 7.79, First post-treatment evaluation = 20.3 ± 8.48, Final post-treatment evaluation = 18.9 ± 8.04 for the control group. (**B**) Change in body image scores compared to baseline score (M ± SEM), as a function of time and treatment. Values (M ± SD) are as follows: First post-treatment evaluation = −3.70 ± 4.71, Final post-treatment evaluation = −5.74 ± 6.63 for the hippotherapy group; First post-treatment evaluation = −0.4233 ± 4.57, Final post-treatment evaluation = −1.83 ± 7.04 for the control group. (***) *p* < 0.001; (**) *p* < 0.01; (*) *p* < 0.05.

**Table 1 cancers-15-01317-t001:** Demographic and clinical features of patients in both groups. BMI: Body Mass Index. Q1: questionnaires fulfilled at the inclusion (baseline); Q2: questionnaires fulfilled one week after the first session of hippotherapy; Q3: questionnaires fulfilled after the last hippotherapy session, six months after the beginning of the program. The quantitative data were statistically compared using a two-sided unpaired Wilcoxon test, and the categorical data using a Chi-squared test; (*) *p* < 0.05; (.) close to significance (between 0.08 and 0.05); (ns) non-significant.

	Hippotherapy Group	Control Group	*p*-Value
Age (years; M ± SD)	52.62 ± 9.93	52.66 ± 9.98	ns (0.7275)
BMI (kg/m^2^; M ± SD)	25.46 ± 5.55	22.22 ± 4.38	ns (0.1137)
Menopausal (%)	53	57	ns (0.506)
Occupation (%): Retired Working life Unknown	17 77 17	9 74 17	ns (0.7207)
Type of breast cancer (%): Ductal carcinoma in situ Invasive ductal carcinoma Lobular carcinoma in situ Invasive lobular carcinoma Triple negative Other	38 54 0 8 8 0	9 78 0 9 0 4	* (0.0265)
Stage(%): T1N0 T2N0 T1N1 T2N1 T2N2	58 4 17 17 4	48 17 17 13 4	ns (0.6839)
Grade (%): 1 2 3	42 38 21	22 48 30	ns (0.3489)
Affected side (%): Left Right	58 42	48 52	ns (0.4441)
Type of treatment (%): Conservative surgery Mastectomy	67 33	91 9	(.) (0.0879)
Radiotherapy Chemotherapy Hormonal therapy	83 50 63	91 52 70	ns (1.0000) ns (0.8349)
Delay (days; M ± SD):			
Q1–Q2	15.0 ± 5.1	12.9 ± 9.3	ns (0.07)
Q2–Q3	201.0 ± 63.4	199.0 ± 44.8	ns (0.22)

**Table 2 cancers-15-01317-t002:** Comparison of questionnaires’ summary scores and subsets obtained at baseline between the two groups at Q1. The total scores (summary scores) and item scores for each clinical test (QoL EORTC QLQ-C30 and QLQ-BR23; cognitive function FACT-Cog V3; fatigue assessment MFI-20; anxiety and depression assessment HADS; body image BIS) are provided as means ± standard deviations. *p*-values correspond to unpaired two-sided Wilcoxon statistical tests. (ns) non-significant; (*) *p* < 0.05.

	Hippotherapy Group	Control Group	*p*-Value
EORTC QLQ-30 (summary score)	70.0 ± 15.0	71.6 ± 11.3	ns (0.956)
Global QoL	57.4 ± 16.5	57.1 ± 14.1	ns (0.654)
Physical functioning	80.0 ±18.6	85.4 ± 12.3	ns (0.379)
Role functioning	67.1 ± 28.0	72.2 ± 27.3	ns (0.416)
Emotional functioning	55.6 ± 25.0	55.1 ± 25.5	ns (0.983)
Cognitive functioning	60.2 ± 24.1	64.1 ± 28.9	ns (0.722)
Social functioning	75.9 ± 28.0	69.2 ± 29.7	ns (0.384)
Symptoms	30.9 ± 15.6	28.3 ± 10.5	ns (0.972)
Fatigue	50.0 ± 23.7	51.0 ± 22.3	ns (0.694)
Nausea/vomiting	6.5 ± 12.8	2.5 ± 6.0	ns (0.262)
Pain	33.8 ± 29.4	29.9 ± 23.4	ns (0.685)
Dyspnea	46.3 ± 61.9	23.5 ± 30.4	* (0.025)
Insomnia	51.9 ± 35.1	51.9 ± 37.4	ns (0.994)
Appetite loss	9.26 ± 23.4	14.8 ± 23.3	ns (0.110)
Constipation	28.7 ± 35.8	30.8 ± 28.2	ns (0.478)
Diarrhea	4.63 ± 11.7	10.3 ± 20.6	ns (0.307)
Financial issues	30.6 ± 35.1	19.2 ± 28.6	ns (0.244)
EORTC QLQ-BR23 (summary score)	65.1 ± 12.8	63.6 ± 10.2	ns (0.568)
Body image	61.3 ± 32.1	59.8 ± 30.9	ns (0.622)
Sexual functioning	77.3 ± 24.6	77.4 ± 21.4	ns (0.897)
Sexual enjoyment	47.1 ± 26.5	50.0 ± 28.6	ns (0.655)
Future perspectives	41.7 ± 30.2	38.1 ± 33.6	ns (0.585)
Side effects from treatment	19.6 ± 13.2	22.8 ± 13.6	ns (0.349)
Breast symptoms	35.4 ± 17.1	40.7 ± 26.6	ns (0.619)
Arm symptoms	25.9 ± 20.4	22.2 ± 19.2	ns (0.558)
FACT-Cog (summary score)	89.3 ± 25.7	88.9 ± 27.4	ns (0.872)
Perceived cognitive abilities	19.2 ± 6.4	18.1 ± 7.7	ns (0.556)
Perceived cognitive impairments	47.1 ± 15.9	47.0 ± 14.9	ns (0.962)
Comments from others	14.4 ± 2.0	14.4 ± 2.7	ns (0.560)
Impact on QoL	7.7 ± 5.5	9.0 ± 4.6	ns (0.329)
IMF (summary score)	59.9 ± 15.1	60.0 ± 13.3	ns (0.940)
General fatigue	14.2 ± 3.6	14.4 ± 3.5	ns (0.924)
Physical fatigue	12.5 ± 4.4	12.5 ± 4.0	ns (0.989)
Mental fatigue	12.1 ± 4.4	11.9 ± 4.3	ns (0.995)
Motivational fatigue	11.1 ± 3.7	10.5 ± 2.0	ns (0.362)
Activity reduction	10.0 ± 3.5	10.8 ± 3.11	ns (0.693)
HADS			
Anxiety	10.0 ± 4.1	10.3 ± 3.8	ns (0.823)
Depression	6.1 ± 4.3	7.0 ± 4.2	ns (0.302)
BIS			
Body image	21.9 ± 7.9	20.7 ± 7.8	ns (0.453)

**Table 3 cancers-15-01317-t003:** General scheme of the conventional supportive care intervention during the six-month follow-up for each patient.

Consulted Specialist	Frequency of Visits
Algologist	Occasionally
Sexologist	One group session
Addictologist	Occasionally
Occupational physician	Occasionally
Homeopath	Every two months
Acupuncturist	Monthly
Physical therapist	Twice a week
Psychologist	Every two weeks
Nutritionist	Two group sessions
Social worker	Occasionally

**Table 4 cancers-15-01317-t004:** Functional items measured and their link to the exercises addressed by the hippotherapy protocol based on work on the horse (movement of the horse at a walk) and by the horse, targeting various interrelated health functioning processes.

Evaluated Items	Contribution through Hippotherapy
**Quality of Life—EORTC QLQ-C30, functional scales**	
Global QoL	- All the hereinafter (2 to 6)
2. Physical functioning	- Reinforcement of global postural balance and fine-tuning of postural responses; - Strengthening of different muscle groups; - Work on upper limbs’ fine motor skills and coordination; - Work on smoothness of movement and psycho-corporal relaxation; - Reinforcement of the body schema and body image; - Breathing techniques and visualizations
3. Role functioning	- Social and cognitive flexibility; - Reinforcement of communication skills; - Stimulation of proactive postures, involvement and motivation; - Self-esteem and self-transcendence
4. Emotional functioning	- Attention to body/emotional feelings; - Reinforcement of the notions of success, pleasure, and disinhibition; - Self-esteem and self-transcendence; - Reinforcement of communication skills; - Linguistic activity of creating meaning (describing, explaining, reappropriating, reinventing, and projecting herself)
5. Cognitive functioning	- Support for attention, concentration, working memory, and executive functions; - Reinforcement of communication skills; - Linguistic activity of creating meaning (describing, explaining, reappropriating, reinventing, and projecting herself)
6. Social functioning	- Reinforcement of the notions of success, pleasure, and disinhibition; - Social and cognitive flexibility; - Reinforcement of communication skills; - Stimulation of proactive postures, involvement and motivation; work on values, meaning, appropriation of symbols
**Quality of Life—EORTC QLQ-C30, symptom scales**	
7. Symptoms	- All the hereinafter (8 to 15)
8. Fatigue	- Work on smoothness of movement and psycho-corporal relaxation; - Breathing techniques and visualizations; - Attention to body/emotional feelings; - Reinforcement of the notions of success, pleasure, and disinhibition; - Management of post-traumatic stress disorder, the grieving process, and the creation of a new life course
9. Nausea/vomiting	- Regulation through movement of bidirectional signaling between ENS and CNS neuromodulatory substances
10. Pain	- Reinforcement of global postural balance and fine-tuning of postural responses; - Work on smoothness of movement and psycho-corporal relaxation; - Reinforcement of the body schema and body image; - Breathing techniques and visualizations; - Attention to body/emotional feelings; - Management of post-traumatic stress disorder;
11. Dyspnea	- Work on smoothness of movement and psycho-corporal relaxation; - Reinforcement of the body schema and body image; - Breathing techniques and visualizations; - Attention to body/emotional feelings; - Management of post-traumatic stress disorder
12. Insomnia	- Work on smoothness of movement and psycho-corporal relaxation; - Breathing techniques and visualizations; - Management of post-traumatic stress disorder
13. Appetite loss	- Regulation through movement of bidirectional signaling between ENS and CNS neuromodulatory substances
14. Constipation	- Regulation through movement of bidirectional signaling between ENS and CNS neuromodulatory substances
15. Diarrhea	- Regulation through movement of bidirectional signaling between ENS and CNS neuromodulatory substances
16. Financial difficulties	- Work on values, meaning, appropriation of symbols; - Linguistic activity of creating meaning (describing, explaining, reappropriating, reinventing, and projecting herself)
**Quality Of Life—EORTC QLQ-BR23, functional scales**	
17. Body image	- Reinforcement of the body schema and body image; - Breathing techniques and visualizations; - Attention to body/emotional feelings; - Self-esteem and self-transcendence
18. Sexual functioning	- Reinforcement of the body schema and body image; - Breathing techniques and visualizations; - Attention to body/emotional feelings; - Self-esteem and self-transcendence
19. Sexual enjoyment	- Reinforcement of the notions of success, pleasure, and disinhibition; - Attention to body/emotional feelings; - Self-esteem and self-transcendence
20. Future perspectives	- Work on values, meaning, appropriation of symbols; - Management of post-traumatic stress disorder, the grieving process, and the creation of a new life course; - Linguistic activity of creating meaning (describing, explaining, reappropriating, reinventing, and projecting herself)
**Quality of Life—EORTC QLQ-BR23, symptom scales**	
21. Side effects from treatment	- Work on smoothness of movement and psycho-corporal relaxation; - Reinforcement of the body schema and body image; - Breathing techniques and visualizations
22. Breast symptoms	- Reinforcement of global postural balance and fine-tuning of postural responses (eyes open and closed); - Strengthening of different muscle groups; - Work on upper limbs’ fine motor skills and coordination; - Work on smoothness of movement and psycho-corporal relaxation
23. Arm symptoms	- Reinforcement of global postural balance and fine-tuning of postural responses (eyes open and closed); - Strengthening of different muscle groups; - Work on upper limbs’ fine motor skills and coordination; - Work on smoothness of movement and psycho-corporal relaxation
**Cognitive functioning—FACT-Cog: memory, verbal fluency, concentration, mental sharpness, resistance to interference, multitasking ability**	
24. Perceived level of cognitive abilities	- Support for attention, concentration, working memory, and executive functions; - Reinforcement of the notions of success, pleasure, and disinhibition; - Social and cognitive flexibility; - Self-esteem and self-transcendence
25. Perceived level of cognitive impairment	- Support for attention, concentration, working memory, and executive functions; - Reinforcement of the notions of success, pleasure, and disinhibition; - Social and cognitive flexibility; - Self-esteem and self-transcendence
26. Impact of the cognitive state on the patient’s quality of life	- Stimulation of proactive postures, involvement and motivation; - Self-esteem and self-transcendence; - Work on values, meaning, appropriation of symbols; - Management of post-traumatic stress disorder, the grieving process, and the creation of a new life course; - Linguistic activity of creating meaning (describing, explaining, reappropriating, reinventing, and projecting herself)
27. Comments from other people on the patient’s cognitive abilities	- Social and cognitive flexibility; - Reinforcement of communication skills; - Self-esteem and self-transcendence
**Fatigue—MFI-20**	
28. General fatigue	- All the hereinafter (29 to 32)
29. Mental fatigue	- Stimulation of proactive postures, involvement and motivation; - Self-esteem and self-transcendence; - Work on values, meaning, appropriation of symbols; - Management of post-traumatic stress disorder, the grieving process, and the creation of a new life course
30. Physical fatigue	- Reinforcement of global postural balance and fine-tuning of postural responses (eyes open and closed); - Strengthening of different muscle groups; - Work on upper limbs’ fine motor skills and coordination; - Work on smoothness of movement and psycho-corporal relaxation;
31. Motivational fatigue	- Reinforcement of the notions of success, pleasure, and disinhibition; - Stimulation of proactive postures, involvement and motivation
32. Activity reduction	- Reinforcement of the notions of success, pleasure, and disinhibition; - Stimulation of proactive postures, involvement and motivation;
**Anxiety and Depression—HADS**	
33. Anxiety	- Work on smoothness of movement and psycho-corporal relaxation; - Breathing techniques and visualizations; - Reinforcement of the body schema and body image; - Self-esteem and self-transcendence; - Management of post-traumatic stress disorder, the grieving process, and the creation of a new life course; - Linguistic activity of creating meaning (describing, explaining, reappropriating, reinventing, and projecting herself)
34. Depression	- Attention to body/emotional feelings; - Reinforcement of the notions of success, pleasure, and disinhibition; - Stimulation of proactive postures, involvement and motivation; - Management of post-traumatic stress disorder, the grieving process, and the creation of a new life course; - Linguistic activity of creating meaning (describing, explaining, reappropriating, reinventing, and projecting herself)
**Body Image—BIS**	
35. Body image	- Work on smoothness of movement and psycho-corporal relaxation; - Reinforcement of the body schema and body image; - Attention to body/emotional feelings; - Reinforcement of the notions of success, pleasure, and disinhibition; - Self-esteem and self-transcendence; - Linguistic activity of creating meaning (describing, explaining, reappropriating, reinventing, and projecting herself)

## Data Availability

The datasets that support the findings in this article are not publicly available due to reasonable privacy and security concerns but are available from the corresponding author upon reasonable request. Access to de-identified data (including data dictionaries) will be available under a data transfer agreement handled by the lead institution (Equiphoria Institute). This will allow the individual participant data underlying the results reported in this article (source data) to be shared after de-identification. The study protocol and the statistical analysis plan can also be made available upon request. Data availability will begin at 9 months and will end at 36 months after publication of the article. Data will be shared according to the following criteria: with investigators who would provide a methodologically sound proposal for analyses that meet the objectives of such approved proposal. The proposal should be addressed to research@equiphoria.com. To access the data, applicants will be required to sign a data access agreement. No source data are provided with this paper taking into account the sensitive nature of health data in addition to the need to regulate the mode of data transfer outside the European Union considering the GDPR standards.

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
