# Peer review of "Breast Cancer: How Hippotherapy Bridges the Gap between Healing and Recovery—A Randomized Controlled Clinical Trial"

_cancers, 2023, doi:10.3390/cancers15041317_

Round 1

Reviewer 1 Report

It is well designed interesting clinical study. The presentation of the manuscript, however can be improved markedly by decreasing remotely relevant information all through the manuscript. Manuscript should be shortened to one third and hippotherapy should be described in technical terms from the beginning of the manuscript including the abstract. Introduction and discussion should be much more specific and should also discuss the possible contribution of physical activity including breathing techniques on the observed effects.  Since these approaches alone may have led to similar outcomes. 

Author Response

Answers from authors

First of all we would like to thank the reviewer for his/her valuable contribution in order to improve the quality of our work.

We have tried to consider how to shorten the paper and moved some information to the Appendices or attached in linked documents. However, the complexity of the clinical trial, its novelty and results as well as the corrections requested by the tree other reviewers made fairly difficult to be more synthetic.

The contribution of physical activity including breathing is indeed commented in the discussion (lines 1049-1059):

“It has been shown that self-induced movement has key effects in the synthesis and release of hormones and peptides strongly involved in brain modulation [94–96]. Exercise has been suggested as the must for improvement of quality of life after breast cancer [97–100]. However, patients are constantly dealing with fatigue, sleeping disorders and pain, which may prevent them from regular voluntary exercise routines. Hippotherapy initiates a physiologic passive movement that rapidly becomes self-induced. This close-to-human gait movement is strongly imprinted in the patient’s body as her own locomotion and rhythm [74]. Within this movement framework, the patient doesn’t feel that she is putting herself in danger and get the space to welcome reward (experiencing pleasure, relaxation, and feeling ready to let go).”

Moreover, even though hippotherapy can be considered as a physical activity, its effects go beyond this issue when taking into account the number of effective days, i.e., 11 days, of this approach during the whole program. Also, even if physical activity and breathing techniques can positively impact the outcome of those patients as reported in several works [1], physical activity programs have failed to enroll and motivate breast cancer survivors due to barriers such as general health issues and lack of time. Also, it has been reported that women may have little confidence in the benefits of physical activity and breast cancer outcomes and consequently, engaging breast cancer survivors in physical activity is currently challenging for health care professionals.

We added a sentence accordingly and the new reference (line 1057) to the article’s bibliography.

We added a technical description of hippotherapy in abstract and introduction:

Abstract (line 30): “Hippotherapy is an emerging specialized rehabilitation approach performed by accredited health professionals and equine specialists on specially trained horses via their movement, activating major paths for physical, mental, psychic and social reinforcement, and been synergistic to rehabilitative and supportive care.”

Introduction (line 96-107): “The horse movement at a walk is biomechanically similar to the human movement when walking and yields to comparable micro-adjustments of the patient’s postural balance, gross and fine motor abilities and reinforcement of sensorimotor control [2]. Additionally, through multimodal inputs (sensory, exteroceptive, proprioceptive, interoceptive, and emotional), hippotherapy has a direct action on the individual’s cognitive potentials and emotional regulation through the interactions of several neural networks [3–10]. During hippotherapy, the specific execution and repetition of a task are key elements of learning/strengthening/promoting a function and a robust rehabilitation and reconstruction process [4,6–11]. It has been reported for example that emotionally empowering multimodal interventions, such as hippotherapy, can provide patients in a late post-stroke phase with life-changing experiences that can have a profound physical and psychological impact [12–14].”

Reviewer 2 Report

Viruega et al reported the results of a randomized clinical trial conducted in a polyclinic setting to test if hippotherapy improves post-treatment quality of life in women with early breast cancer. They utilized standard instruments to measure QoL trends up to 6 months after curative cancer treatment. The results indicate a significant and lasting improvement of QoL in groups receiving hippotherapy.

In general, the rigor and quality of this research is comparable to commonly mentioned alternative medicine research in terms of QoL improvement for women with early breast cancer (Ref: Vadiraja et al. Complementary Therapies in Medicine 17:274-280, 2009). Despite the inherent limit of alternative interventions such as heterogenous treatment intensity only described qualitatively and placebo effects, the data supported a likely meaningful improvement for primary endpoints. However, the drop out rate in the patients receiving multimodality (surgery+2 adjuvant treatment) treatments randomized to control group is a significant concern when interpreting the results. Description and analysis of the clinical factors (types of surgery, stage of the disease, presence of chemotherapy, radiotherapy, or hormone therapy) is very limited, and that interferes with evaluation when comparing to the existing literature of QoL improvement with alternative medicine approached in women with early breast cancer.

Specific comments

1. Please describe more about the ethic review committee as it is outside an university and very limited information can be found regarding its process and quality of ethics review.

2. The authors need to provide disease demographics according to common standards in breast cancer research, such as disease stages, types of surgery, biological subtypes, presence of chemotherapy, radiotherapy, and hormone therapy, and treatment non-compliance rate. 

3. Potential conflicts of interest with hippotherapy providers and research sponsors need to be declared. 

4. The sexual functioning scores in QLQ-BR23 reported in this paper significantly differed from European reference value (ref: Karsten et al. European Journal of Cancer 163:128-139, 2022). Please elaborate on reasons for this significant difference.

5. The authors are encouraged to discuss more on the interfernce from placebo effects in this open label clinical trials. Especially how do trial participants value hippotherapy as a luxurious activity and how this contributes to observed QoL change need to be discussed in the psychosocial context.

Author Response

Answers from authors

First of all we would like to thank the reviewer for his/her valuable contribution in order to improve the quality of our work.

However, the dropout rate in the patients receiving multimodality (surgery+2 adjuvant treatment) treatments randomized to control group is a significant concern when interpreting the results. Description and analysis of the clinical factors (types of surgery, stage of the disease, presence of chemotherapy, radiotherapy, or hormone therapy) is very limited, and that interferes with evaluation when comparing to the existing literature of QoL improvement with alternative medicine approached in women with early breast cancer.

We came back to the original clinical files of our patients and better detailed the type of treatment without taking into account the stratified randomization categories (see new Table 1). Both populations are actually comparable with respect to radiotherapy, chemotherapy and hormonal therapy.

We consequently deleted the sentence in Study Limitations and Perspectives (line 1181):

“In the case of our study, however, we observed a better response of the hippotherapy group despite a higher frequency of chemotherapy-treated patients in this group. This might suggest a strong effect of hippotherapy on the recovery of heavily impacted patients.”

  1. Please describe more about the ethic review committee as it is outside an university and very limited information can be found regarding its process and quality of ethics review.

We added a short explanation on the Ethical Review procedure in France (line 134).

“In France, the Jardé’s law defines Research Involving the Human Person (RIPH) as research organized and carried out on healthy or sick volunteers, with a view to developing biological or medical knowledge and which aims to evaluate safety and efficacy of the proposed healthcare solution. RIPH may only be carried out after a favorable opinion from a university hospital ethical committee (CPP) drawn at random at national level from an IT platform set up by the French health authorities. This new procedure replaces the former procedure of ethical evaluation by local university committees.”

  1. The authors need to provide disease demographics according to common standards in breast cancer research, such as disease stages, types of surgery, biological subtypes, presence of chemotherapy, radiotherapy, and hormone therapy, and treatment non-compliance rate. 

We provided a disease demographics in Table 1 according to common standards in breast cancer (tumor subtype, grade, stage, side, treatment; see Table 1).

  1. Potential conflicts of interest with hippotherapy providers and research sponsors need to be declared. 

We provided potential conflict of interest (line 1228):

“H.V. has shares in the Equiphoria Institute and Alliance Equiphoria. The remaining authors declare no competing interests.”

  1. The sexual functioning scores in QLQ-BR23 reported in this paper significantly differed from European reference value (ref: Karsten et al. European Journal of Cancer 163:128-139, 2022). Please elaborate on reasons for this significant difference.

Sexual functioning scores similar to ours have been reported in other European [15–18] and non-European [19–22] clinical trials. Also, even lower scores with respect to Karsten’s work [23] have been reported in other European trials [24–26]. The sexual sphere potentially depends on multiple variables and confounding factors (age, hormonal status, cultural and contextual issues, kind of primary therapy, etc.). Due to the noticed variability among studies, it is fundamental and mandatory to have a comparable baseline between the trial groups in order to prevent major biases. In our case, the sexual functioning scores were comparable but high which is probably a disadvantage since it is theoretically difficult to reveal a significant difference between both groups due to a too low delta.

However, since the sexual functioning remained comparable in both cohorts, we didn’t comment on this issue in the text but add the Karsten’s reference to the bibliography.

  1. The authors are encouraged to discuss more on the interference from placebo effects in this open label clinical trials. Especially how do trial participants value hippotherapy as a luxurious activity and how this contributes to observed QoL change need to be discussed in the psychosocial context.

Even though placebo effect is a reality in cancer clinical trials, both positive and negative effects have been described. Around third of the patients of a controlled trial can show positive placebo effects. On the other hand, nearly a quarter of patients taking a placebo experience side effects (nocebo effects) [27]. In the context of the trial, hippotherapy is definitely not associated to a luxurious activity since the care is carried out in a rigorous medicalized environment of a specialized hippotherapy center exclusively implemented to this end at the opposite of an equestrian luxurious environment where social and entertainment standards are given to the users.

We added the above paragraph to the Study Limitations and Perspectives (line 1170).

Reviewer 3 Report

Cancer 2181862

The primary inclusion criteria includes:

The 6-months program included, in the treated group, an initial 1-week daily hippotherapy session followed by three short 2-days sessions with an interval of 2 months between each where the patients received conventional supportive care. The correct forms are “6-month” and “2-day”, please check the grammar throughout the manuscript (I have found some issues).

The abstract is generally well structured.

The introduction is generally well-written, however, the readers would benefit from more detailed explanation about the hippotherapy programme and also the 3 x 2 day sessions and what exactly these sessions consisted of. This will allow easier replication of this programme by other studies. 

Also, some evidence about how these two sets of treatments improved the patients' well-being and quality of life (and other mentioned criteria). I appreciate that you have mentioned various papers, but it wouldn’t hurt to give an example to expand on the links. 

Also, since aspects such as stage and TNM are referred to in the rest of the paper, the authors could define these in the introduction and any relevant breast cancer subtypes information that may be referred to later. 

Patients were excluded whether they had a history of malignancies in the last 5 years with the exception of basal cell skin carcinoma or squamous cell carcinoma, a history of chronic fatigue syndrome, concomitant and uncontrolled severe degenerative or chronic diseases, clinically significant cognitive impairment or dementia, pregnancy and breastfeeding, history of therapeutic horse riding or hippotherapy during the last 6 months, and whether the patient was participating in another biomedical research or in exclusion period. I think the word ‘whether’ should be replaced by if. Also, the list of exclusion criteria is slightly strange. For example, why were skin malignancies specifically mentioned in this exclusion criteria?

About table 1, did the breast cancer subtype also make a difference in this selection?

The minimum reasonably significant difference was estimated to be 10 points for the primary outcome (EORTC QLQ-C30) [27–29] and the standard deviation σ = 16.44. This aspect should be explained more.

Six semi-quantitative self-administered questionnaires (i.e., EORTC QLQ-C30 and QLQ-BR23, FACT-Cog, MFI-20, HADS, BIS; see below for explanation) were completed by the patients. I assume the authors have included this information (detailed questionnaire content) either in a supplementary or in a valid link?

Baseline correction was achievable since baseline values were comparable in the two groups. Please provide more detail about base line correction. An outsider is not going to know how you have performed this.

We tested the normality of the data using a Shapiro-Wilk test and none of the data followed a normal distribution, requiring us to use nonparametric statistics. Why were the data obtained nonparametric?

In table 2 please provide detail about some of the acronyms provided and explain them. Also, in line 144 when the authors talk about the table, please explain some of the aspects mentioned in the table, also in the main text. There currently is a disconnect.

 In table 2 it is not clear to me how the scores of say 77.5 and 73.2 in hippo and control groups were set. Did the questionnaires come with per-question scores?

In line 166, the standard deviation is more accurate than the standard error of the mean.

before the healthcare program (baseline evaluation), after the first session of one week of the healthcare program (first post-treatment evaluation), and at the end of the six-month healthcare program (final evaluation). Why was the second questionnaire set so early into the programme?

How practical and feasible is hippotherapy to the average French person, especially those living in locations which may be far from these facilities?

The topics in lines 187- 200, although seemingly in good detail could be provided in a link for easier replication (e.g., what exact exercises were intended) hoping to use your methods and protocols for their own studies.

The protocol timeline was optimized considering the potential strong imprint brought during the initial session. What does this mean? Strong imprint?

In line 208, MIS need not be spelt out again.

In figure one, please make it more clear what treatments the control group received. I read this as only on conventional treatment on week 28 and obviously no hippo, is that correct? Wouldn’t it be better, if the control gets exactly the same as the hippo group minus the hippotherapy? Since now the control group are only getting one conventional treatment on week 28, so there is a lower frequency of intervention for them which is another variable (unless I am misunderstanding your figure).

Please note that the acronyms of the questionnaires only appear in lines 237-242, even though they appear as acronyms well before this.

As mentioned kindly make the exact content of the questionnaires assessable to all, as part of the data associated with this study.

The formulas in lines 265-270 are useful, thanks. I commented earlier that I don’t understand the scores in table 2. This explanation here does make it clearer but since the reader won’t have known this when they read table 2, it would be good to perhaps guide the readers to where to look.

The selection of the type of questionnaires was logical and well-thought-out.

Except for one symptom item of the EORTC QLQ-C30 (the dyspnea symptom). The authors could explain why this is the case.

Figure 2, is it necessary to show both A graphics, from what I understand you are showing the raw data in the left-hand graph and the query minus baseline in the second. If this is correct, please only keep the right-hand one (this applies to all the other figures). Also, any entity added to a figure must have its own annotation (hence me referring to A left and right hand since they can’t be referred to any other way).

In figure 2B, the upper panel needs the titles as well (global QoL, physical etc), this makes it easier for the reader to follow. The data is good, thanks.

It is very promising that the authors have tried to quantify, normalise and perform statistical tests on these fairly subjective questionnaires.  Many such studies, leave it at a descriptive level and don’t bother. Thanks for this.

Figures 3-6 also show interesting and promising differences between the study cohorts and I think this study has been very useful. Also, the results have been described in good detail with ample statistical depth.

A stylistic issue with figure 4-6, why have figure A (right) and B graphs been inverted?

Author Response

Answers from authors

First of all we would like to thank the reviewer for his/her valuable contribution in order to improve the quality of our work.

The 6-months program included, in the treated group, an initial 1-week daily hippotherapy session followed by three short 2-days sessions with an interval of 2 months between each where the patients received conventional supportive care. The correct forms are “6-month” and “2-day”, please check the grammar throughout the manuscript (I have found some issues).

We checked grammar issues according to the comments.

The abstract is generally well structured.

Although the abstract is well structured, we improved it according to the comments of the other reviewers.

The introduction is generally well-written, however, the readers would benefit from more detailed explanation about the hippotherapy programme and also the 3 x 2 day sessions and what exactly these sessions consisted of. This will allow easier replication of this programme by other studies. Also, some evidence about how these two sets of treatments improved the patients' well-being and quality of life (and other mentioned criteria). I appreciate that you have mentioned various papers, but it wouldn’t hurt to give an example to expand on the links. 

We better explained the hippotherapy program in the introduction in order to allow easier replication by others. We also added an example to expand on the links (line 104):

“It has been reported for example that emotionally empowering multimodal interventions, such as hippotherapy, can provide patients in a late post-stroke phase with life-changing experiences that can have a profound physical and psychological impact [12–14].”

Also, since aspects such as stage and TNM are referred to in the rest of the paper, the authors could define these in the introduction and any relevant breast cancer subtypes information that may be referred to later. 

We added a sentence to better define the cohort in the introduction (line 110):

"The cohort consisted of women diagnosed with stage IIIA breast cancer who had completed primary treatment (surgery, radiotherapy, chemotherapy) with the exception of hormonal therapy.”

Moreover, we better elaborated on the relevant disease demographics (see Table 1) according to common standards in breast cancer (tumor subtype, grade, stage, side, treatment).

Patients were excluded whether they had a history of malignancies in the last 5 years with the exception of basal cell skin carcinoma or squamous cell carcinoma, a history of chronic fatigue syndrome, concomitant and uncontrolled severe degenerative or chronic diseases, clinically significant cognitive impairment or dementia, pregnancy and breastfeeding, history of therapeutic horse riding or hippotherapy during the last 6 months, and whether the patient was participating in another biomedical research or in exclusion period. I think the word ‘whether’ should be replaced by if. Also, the list of exclusion criteria is slightly strange. For example, why were skin malignancies specifically mentioned in this exclusion criteria?

We changed the word “whether” by “if” as recommended (exclusion criteria). We clarified the exclusion criterion.

About table 1, did the breast cancer subtype also make a difference in this selection?

Table 1: We introduced statistics for cancer subtypes and other disease demographics.

The minimum reasonably significant difference was estimated to be 10 points for the primary outcome (EORTC QLQ-C30) [27–29] and the standard deviation σ = 16.44. This aspect should be explained more.

We better explained the sample size estimation and corrected a typo error for the reference and value in the variance (line 176):

“Sample size was determined to detect a difference with a power of 80% using a two-sample t-test at a two-sided significance level of 5%. The minimum reasonably significant difference retained for the primary outcome (EORTC QLQ-C30) was Δ = 10 points according to other similar studies [28–30]. The variance in total QoL in four recently published studies [30–33] is respectively σ = 13.9 (n = 19), σ = 17.8 (n = 33), σ = 9.1 (n = 75) and σ = 16.4 (n = 25). We chose the study of Vardar [32], which seemed the most adapted to the characteristics of our population and which showed an important dispersion in a small cohort. The standard deviation used to calculate the sample size was therefore σ = 13.9.”

Six semi-quantitative self-administered questionnaires (i.e., EORTC QLQ-C30 and QLQ-BR23, FACT-Cog, MFI-20, HADS, BIS; see below for explanation) were completed by the patients. I assume the authors have included this information (detailed questionnaire content) either in a supplementary or in a valid link?

We included the CRF containing the exact content of the questionnaires assessable to all in a document through a link (see HippoBreastCa_CRF_VE_31012017.pdf).

Baseline correction was achievable since baseline values were comparable in the two groups. Please provide more detail about base line correction. An outsider is not going to know how you have performed this.

We provided more details about baseline correction (line 203), and corrected the graph inversion:

“Baseline correction was achieved by subtracting the score obtained at Q1 from score obtained at Q2 and Q3 for each subset score and for each participant. This allowed us to set all the participants to a score of zero and erase the variability of the baseline states. Also, by doing so, we were able to compare the differences in score changes between the two groups. When baseline correction is performed, a positive score observed at Q2 or Q3 means an increase of score compared to Q1, and a negative score observed at Q2 or Q3 implies a decrease of score compared to Q1.”

We tested the normality of the data using a Shapiro-Wilk test and none of the data followed a normal distribution, requiring us to use nonparametric statistics. Why were the data obtained nonparametric?

We clarified the non-normality of data (line 216):

“We tested the normality of the data using a Shapiro-Wilk test and none of the data followed a normal distribution. This was predictable as none of the score was following a continuous distribution. In order to be as rigorous as possible, considering the non-continuous distribution, the relative small sample sizes and the results of the Shapiro-Wilk test, we decided to use nonparametric statistics.”

In table 2 please provide detail about some of the acronyms provided and explain them. Also, in line 144 when the authors talk about the table, please explain some of the aspects mentioned in the table, also in the main text. There currently is a disconnect.

We clarified the acronyms of Table 2 in the legend:

“Table 2. Comparison of questionnaires’ summary scores and subsets obtained at baseline between the two groups at Q1. The total scores (summary scores) and item scores for each clinical test (QoL EORTC QLQ-C30 and QLQ-BR23; cognitive function FACT-Cog V3; fatigue assessment MFI-20; anxiety and depression assessment HADS; and body image BIS) are provided as means ± standard deviations. P values correspond to unpaired two-sided Wilcoxon statistics. (ns) non-significant; (*) p<0.05.”

We explained some aspects mentioned in the table in the main text. The summary scores for EORTC QLQ-C30 and EORTC QLQ-BR23 are explained in the text (line 409). We clarified that and referred to text (line 210):

“Baseline correction was achievable since baseline values were comparable in the two groups (no significant differences). Table 2 shows the summary scores (see “2.4.1. Primary end point” for the computing formula of EORTC QLQ-C30 and EORTC QLQ-BR23) of the different clinical tests as well as the individual scores for each item before the beginning of the program for the hippotherapy and the control group, and the statistical result of the two-by-two comparison.”

In line 166, the standard deviation is more accurate than the standard error of the mean.

We chose SEM instead of SD in the Figures only for graphical reasons knowing that standard deviation is more accurate. To complete the presentation of data, we added tables (Tables A1 and A2) in Appendix B with the values as Means ± Standard deviations (comparisons of both groups for Q2 and Q3). We added an explanation of the choice of SEM in the Figures (line 265):

“We choose SEM to display on the figures for graphical reasons. However, the standard deviations (SD) are provided in Table 2 and Appendix B Table A1 and Table A2 for more accuracy.”

… before the healthcare program (baseline evaluation), after the first session of one week of the healthcare program (first post-treatment evaluation), and at the end of the six-month healthcare program (final evaluation). Why was the second questionnaire set so early into the programme?

The protocol timeline was optimized considering the potential strong imprint brought during the initial session. What does this mean? Strong imprint?

We better explained the reasons for the earliness of Q2 which answers also the following point concerning the timeline and the “strong imprint” (line 317):

“It is challenging to determine the minimum effective quantity of a required rehabilitation or supportive care. Scientific evidence is lacking and studies have biases of several kinds. To date, the effectiveness of many such techniques has not been systematically demonstrated. A hippotherapy session lasts one hour per day during which between 3000 and 4500 contractions of each postural muscle are sequentially realized in a background mode (horse at a walk) in parallel to other requests (fine motor skills, cognitive elaboration, and psychic work), well beyond what a conventional rehabilitation or supportive care session allows. Given the intensity of each session that mobilizes the individual in his/her entirety (e.g., somatic, sensory, cognitive, emotional and motivational, and psychic spheres), and relying on the enriched environment brought by hippotherapy, we have most of the time noticed a remarkable prompt functional improvement, even beyond the theoretical period of consolidation of an outcome after the beginning of a disease. Naturally, we unequivocally respect a certain rhythm by integrating the duration of the patient’s processing of physical/mental skills and the ensuing fatigue. Overall, the strong stimulation of the sensory, sensitive, and motor spheres promotes and interacts with the mechanisms related to the tasks’ performances in the cognitive and emotional domains through the activation of multiple neural networks [34,35]. The degree of change associated with neuroplasticity through hippotherapy is most likely linked to both the relevance of the activity and the intensity and frequency of the elements that constitute it [36–38] and constitutes a strong imprinting for functional and behavioral remodeling.”

How practical and feasible is hippotherapy to the average French person, especially those living in locations which may be far from these facilities?

Significant efforts have been made to effectively disseminate the hippotherapy approach among key opinion leaders and the scientific and medical communities, resulting in a steadily increasing volume of patients from across France during the last decade. Care through hippotherapy (rehabilitation, neurorehabilitation, supportive care) is still considered by the French social security as an additional non-refundable cost. Nevertheless, the timely and sustainable medical results obtained with this approach in the last decade (published clinical studies) has encouraged many French national insurance companies to reimburse up to 100% of the program costs (all-inclusive) to patients. From the practical point of view, the stay is organized on site during the time of intervention, and transport, accommodation and other practical aspects are scheduled by the institute in advance according to the patient’s needs. Travelling issues are not considered by the patients as insurmountable constraints and often the possibility of an elsewhere decontextualized from the everyday is very positively perceived.

We didn’t add this information to the article since it can be misjudged as a business strategy, which is not.

The topics in lines 187- 200, although seemingly in good detail could be provided in a link for easier replication (e.g., what exact exercises were intended) hoping to use your methods and protocols for their own studies.

We added the most common exercises used during the program (line 311):

“A routine of exercises was used according to the needs of each patient on the horse (breathing exercises; visualizations; body scan; relaxation exercises; active movement of different body segments and exercises on body contextualization; therapeutic vaulting) and by the horse (choice of the horse and grooming; various walking courses with the horse; round pen work; blindfolded guidance or grooming; work at liberty).”

In line 208, MIS need not be spelt out again.

We deleted Montpellier Breast Institute and kept MIS from line 342 accordingly.

In figure one, please make it more clear what treatments the control group received. I read this as only on conventional treatment on week 28 and obviously no hippo, is that correct? Wouldn’t it be better, if the control gets exactly the same as the hippo group minus the hippotherapy? Since now the control group are only getting one conventional treatment on week 28, so there is a lower frequency of intervention for them which is another variable (unless I am misunderstanding your figure).

We modified the Figure 1 and clarified the legend:

“Figure 1. Timeline of the clinical trial: conventional support care was provided to the control group during 28 weeks whereas the hippotherapy group received the conventional support care for the same time minus the days where they received the hippotherapy treatment. The aim of such design was to prevent an unequal amount of care among groups.”

Please note that the acronyms of the questionnaires only appear in lines 237-242, even though they appear as acronyms well before this.

We moved the acronyms’ explanation to their first citing location (line 193) in the text and deleted from line 372:

“Six semi-quantitative self-administered questionnaires were completed by the patients:

  • The European Organization for Research and Treatment of Cancer quality of life questionnaire (EORTC QLQ-C30 and EORTC QLQ-BR23)
  • The Functional Assessment of Cancer Therapy-Cognitive Function (FACT-Cog V3)
  • The Multidimensional Fatigue Inventory (MFI-20)
  • The Hospital Anxiety and Depression (HADS)
  • The Body Image Scale (BIS)

The formulas in lines 265-270 are useful, thanks. I commented earlier that I don’t understand the scores in table 2. This explanation here does make it clearer but since the reader won’t have known this when they read table 2, it would be good to perhaps guide the readers to where to look.

We guided the reader to the formulas (line 212).

The selection of the type of questionnaires was logical and well-thought-out.

We selected the questionnaires according to the literature about clinical trials in the field, thank you

Except for one symptom item of the EORTC QLQ-C30 (the dyspnea symptom). The authors could explain why this is the case.

We discussed the dyspnea difference between both groups in the baseline data (line 471).

“In this regard, all patients in the trial were assessed for cardiac function. None of the patients in any group showed evidence of cardiac dysfunction. Although dyspnea values were significantly higher in the hippotherapy group at baseline, they decreased as early as one week after the start of the trial (see Appendix B; Table A1 and Table A2), and so, the initial difference was not considered relevant.”

Figure 2, is it necessary to show both A graphics, from what I understand you are showing the raw data in the left-hand graph and the query minus baseline in the second. If this is correct, please only keep the right-hand one (this applies to all the other figures). Also, any entity added to a figure must have its own annotation (hence me referring to A left and right hand since they can’t be referred to any other way).

In figure 2B, the upper panel needs the titles as well (global QoL, physical, etc.), this makes it easier for the reader to follow. The data is good, thanks.

We improved Figures (labels, titles, formats) and legends. We decided to keep panels A and B despite the reviewer’s comment since doing so permitted first to report whether the two groups showed an improvement across the treatment period. Also, we could observe if these improvements appeared quickly after the beginning of the treatment (Q2) or later (Q3) and if they are maintained across time. On the other hand, the reader can easily be aware of the differences in score changes between the two groups in relative values.

A stylistic issue with figure 4-6, why have figure A (right) and B graphs been inverted?

We corrected the inversions in Figures 4-6 for a better homogeneity in graphical reporting.

Reviewer 4 Report

In this study, the Authors presented new data indicating that hippotherapy can be the key initial support care of integrative therapy in breast cancer patients. The investigation of  thirty-five various parameters covered by six clinical scales evaluating different functions related to patient’s self-perception  indicated that hippotherapy improves self esteem, cognitive abilities, physical and  emotional states and brain-gut relationships coordinated.

This study provides new important evidence that can be used as a productive strategy of patient rehabilitation after breast cancer treatment.

There are just small concerns:

1. Abstract: Please, underline the most important and novel findings in more details.

2 One of the very important points is Dyspnea. It can be caused by cancer treatment associated cardiac complications. Please, specify, did you evaluated cardiac functions before hippotherapy prescription?  It would be also good to underline positive effects of hippotherapy on Dyspnea.

3. Do you know something about the effects of hippotherapy on blood cortisol level? If yes, it would be good to share with this knowledge because cortisol impairs immunity.

4. If it is possible, please, prepare one more Table indicating the same parameters as the Table 2  but for the long-tern hippotherapy treatment (i.e. for Q2 and/or Q3).

Author Response

Answers from authors

First of all we would like to thank the reviewer for his/her valuable contribution in order to improve the quality of our work.

  1. Abstract: Please, underline the most important and novel findings in more details.

We underlined the most important findings in the abstract (line 43):

“We observed statistical differences in the evolution of the measured parameters over time between the two groups. The hippotherapy group showed a much faster, favorable and continuous improvement until the end of the program for each function assessed. The most striking improvements were in global quality of life, and fatigue, while breast cancer-specific quality of life, cognitive performance, anxiety and depression and body image showed a less marked but still statistically significant difference at the final post-treatment evaluation.”

  1. One of the very important points is Dyspnea. It can be caused by cancer treatment associated cardiac complications. Please, specify, did you evaluated cardiac functions before hippotherapy prescription?  It would be also good to underline positive effects of hippotherapy on Dyspnea.

We added information about evaluation of the cardiac function prior to inclusion in the trial and reported the normality of the cardiac function. In this circumstances it is difficult to explain the slight (and unique ) statistical difference among the two groups. This difference disappeared from the second evaluation, one week after the beginning of the trial (decrease of dyspnea at Q2 and Q3, see Appendix B; Tables A1 and A2 in Appendix B). We added the following information in line 471:

“In this regard, all patients in the trial were assessed for cardiac function. None of the patients in any group showed evidence of cardiac dysfunction. Although dyspnea values were significantly higher in the hippotherapy group at baseline, they decreased as early as one week after the start of the trial (see Appendix B; Table A1 and Table A2), and so, the initial difference was not considered relevant.”

  1. Do you know something about the effects of hippotherapy on blood cortisol level? If yes, it would be good to share with this knowledge because cortisol impairs immunity.

We added in the discussion about molecular changes elicited by hippotherapy the unique paper about hippotherapy and decrease of cortisol levels in ASD child (line 1086):

“Finally, decrease of some key hormones related to stress and immune response has been suggested after hippotherapy [39].”

  1. If it is possible, please, prepare one more Table indicating the same parameters as the Table 2  but for the long-tern hippotherapy treatment (i.e. for Q2 and/or Q3).

We added two supplementary tables in Appendix B (Tables A1 and A2) with means and standard deviation for each item evaluated corresponding to Q2 and Q3 (same as Table 2).

Round 2

Reviewer 1 Report

Required changes were included in the revised manuscript. 

Author Response

Thank you very much for your review and advice.

Reviewer 2 Report

The comments are adequately addressed. Congratulations.

Author Response

Thank you for your review.

Reviewer 3 Report

The authors have addressed my comments

Author Response

Thank you for your review.

Reviewer 4 Report

The Authors addressed all my concerns. The Article can be published.

I am very grateful to the Authors because they uncovered beauty and strength of hippothreapy. 

The text of this article is definitely healing. 

Author Response

Thank you for your review.